# DSPO: Direct Score Preference Optimization for Diffusion Model Alignment

**Huaisheng Zhu, Teng Xiao & Vasant G Honavar**
Pennsylvania State University
{`hvz5312,tengxiao,vhonavar`}@psu.edu

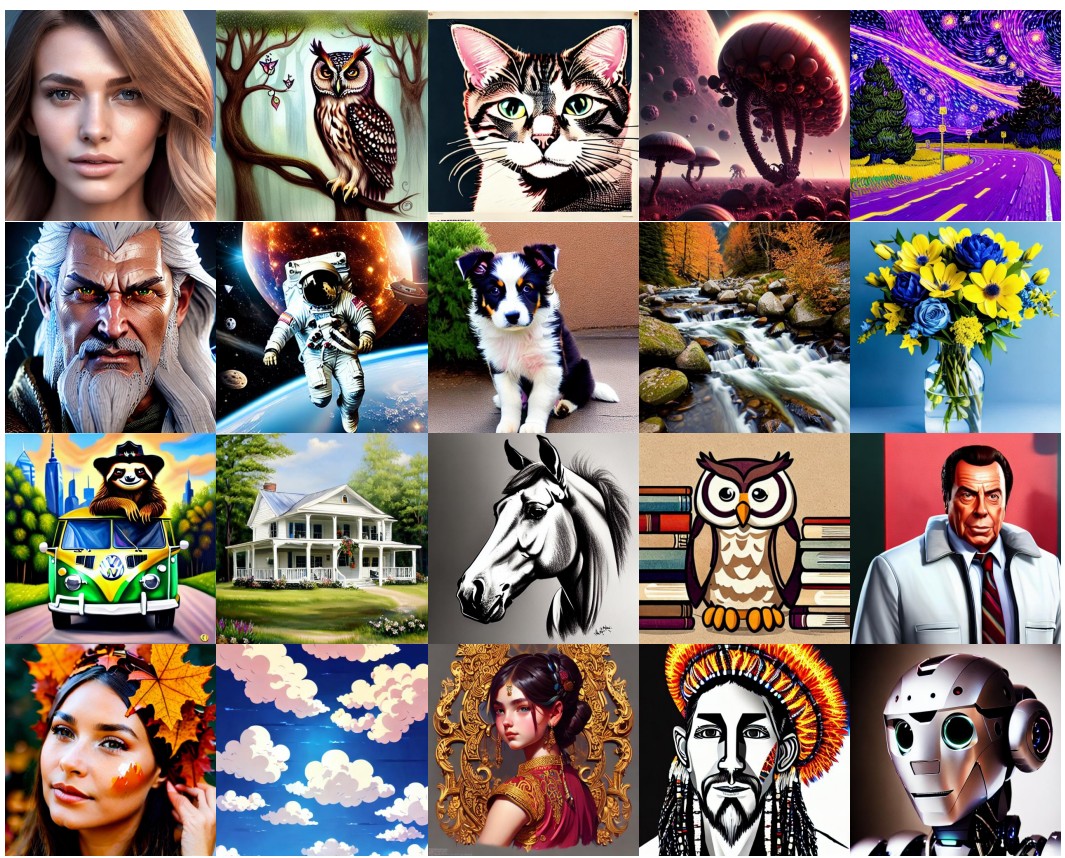

Figure 1: Sample images generated by Stable Diffusion v1.5, fine-tuned using our proposed Direct Score Preference Optimization (DSPO). DSPO aligns human preferences of images through preference score matching, maintaining consistency with the pretraining objective. With DSPO fine-tuning, Stable Diffusion v1.5 produces high-quality images that not only adhere more closely to the text prompts but are also visually striking and more appealing.

## Abstract

Diffusion-based Text-to-Image (T2I) models have achieved impressive success in generating high-quality images from textual prompts. While large language models (LLMs) effectively leverage Direct Preference Optimization (DPO) for fine-tuning on human preference data without the need for reward models, diffusion models have not been extensively explored in this area. Current preference learning methods applied to T2I diffusion models immediately adapt existing techniques from LLMs. However, this direct adaptation introduces an estimated loss specific to T2I diffusion models. This estimation can potentially lead to suboptimal performance through our empirical results. In this work, we propose Direct Score Preference Optimization (DSPO), a novel algorithm that aligns the pretraining and fine-tuning objectives of diffusion models by leveraging score

matching, the same objective used during pretraining. It introduces a new perspective on preference learning for diffusion models. Specifically, DSPO distills the score function of human-preferred image distributions into pretrained diffusion models, fine-tuning the model to generate outputs that align with human preferences. We theoretically show that DSPO shares the same optimization direction as reinforcement learning algorithms in diffusion models under certain conditions. Our experimental results demonstrate that DSPO outperforms preference learning baselines for T2I diffusion models in human preference evaluation tasks and enhances both visual appeal and prompt alignment of generated images. The source code for DSPO is publicly available at the Github: https://github.com/huaishengzhu/DSPO.

# 1 INTRODUCTION

Diffusion-based Text-to-Image (T2I) models have achieved remarkable success in generating high-quality images from textual prompts (Ramesh et al., 2021; Saharia et al., 2022; Rombach et al., 2022). These models are generally trained in a single stage, utilizing web-scale datasets of text-image pairs and employing the diffusion objective to guide the learning process. While large language models (LLMs) have made substantial progress in generating text that addresses a wide array of human needs, they achieve this through a two-step process: pretraining on vast, noisy datasets from the web, followed by fine-tuning on smaller, more specific datasets to align with user preferences (Achiam et al., 2023; Dubey et al., 2024). This fine-tuning phase refines the model's outputs to better meet human expectations, without significantly compromising the broader capabilities gained during pretraining. Applying this fine-tuning approach to text-to-image models could similarly enhance image generation in line with user preferences—an area that, to date, has been relatively underexplored compared to advancements in the language domain.

Several recent studies have focused on fine-tuning diffusion-based T2I models to better align with human preferences after large-scale pretraining, which is often achieved through Reinforcement Learning from Human Feedback (RLHF) (Black et al., 2023; Clark et al., 2023; Fan et al., 2024; Lee et al., 2023; Prabhudesai et al., 2023; Uehara et al., 2024). These approaches typically involve fitting a reward model to a dataset of human preferences and optimizing the diffusion model to generate images that receive high reward scores, while avoiding significant deviation from the original model. However, building a reliable reward model for diverse tasks poses challenges, often requiring a large collection of images and substantial training resources (Wallace et al., 2024; Rafailov et al., 2024).

To address this issue, several recent works (Wallace et al., 2024; Yang et al., 2024; Li et al., 2024; Gu et al., 2024) have introduced preference learning methods that eliminate the need of reward models when fine-tuning diffusion models for human preferences inspired by the success of Direct Preference Optimization (DPO) (Rafailov et al., 2024). These approaches directly adapt the objectives used in LLMs for human preference alignment to diffusion models, adjusting them to fit the specific formulation of diffusion models. Immediate adaptation results in an estimated loss on diffusion models based on the original DPO objectives. For example, the loss of Diffusion-DPO is upper-bounded by the original DPO loss (Wallace et al., 2024).

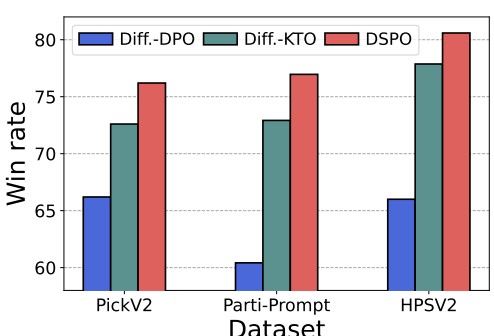

Figure 2: Win-rate (vs SD15) for DSPO and preference learning baselines based on Aesthetics reward. "Diff." represents "Diffusion".

This estimation may result in suboptimal performance when fine-tuning diffusion models for human preference alignment through empirical results, as demonstrated in Figure 2. The figure shows results from human preference alignment experiments on three widely used datasets, comparing DSPO with existing baselines for preference learning in T2I diffusion models.

In this paper, we introduce a new perspective about how to fine-tune diffusion models by aligning their output distribution with human preferences through score matching, the same technique used in pretraining. It is known that diffusion models can be formulated as stochastic differential

equations (SDEs) and are trained using score matching objectives, which is why they are also referred to as score functions or score models (Song et al., 2020). Based on this formulation, we propose Direct Score Preference Optimization (DSPO), an algorithm that distills the score function of human-preferred image distributions into pretrained score functions (diffusion models). Here, we introduce the target human-preferred score function by combining the ground-truth score of the data distribution with the score from a theoretical preference model, such as the Bradley-Terry model (Bradley & Terry, 1952). To simplify this process, we incorporate the implicit reward formulation from DPO, eliminating the need for additional training of the preference model. This target score function models the image distribution aligned with human preferences, and fine-tuning the pretrained score models to match this target guides the diffusion process toward human-preferred outputs. Furthermore, we theoretically demonstrate that the optimization direction of this preference score matching loss (under some conditions) is equivalent to the direction required to optimize the RLHF objective introduced in diffusion models using reinforcement learning.

The main contributions of this paper are: (i) To the best of our knowledge, we are the first to fine-tune diffusion models based on human preferences using a score-matching approach that aligns the pretraining and fine-tuning objectives. This introduces a novel perspective for designing preference learning algorithms for diffusion models. (ii) we theoretically prove that DSPO shares the same optimization direction with RLHF objectives in diffusion models under certrain conditions. (iii) DSPO outperforms preference learning baselines on evaluations of human preference tasks.

## 2 RELATED WORKS

**Text-to-image Diffusion Models.** Denoising diffusion probabilistic models have proven to be powerful tools for generating diverse data types (Ho et al., 2020). Additionally, the sampling process of diffusion models can be interpreted as stochastic differential equations (SDEs) and is trained using score matching objectives based on this formulation (Song et al., 2020). These models have been successfully applied in various fields, including image synthesis, video generation, and robotics control. Notably, text-to-image diffusion models have enabled the generation of highly realistic images from textual descriptions, paving the way for new possibilities in digital art and design (Ramesh et al., 2021; Saharia et al., 2022). Recent research has focused on improving the control and precision of diffusion models during the generative process. Techniques such as adapters and compositional approaches have been introduced to incorporate additional constraints and blend multiple models, enhancing both image quality and generation control (Zhang et al., 2023; Du et al., 2023). Additionally, classifier-based and classifier-free guidance methods have significantly advanced the autonomy of diffusion models (Dhariwal & Nichol, 2021; Ho & Salimans, 2022), allowing them to generate outputs that closely align with user intentions. In our work, we adopt Stable Diffusion (Rombach et al., 2022) to generate images based on specific textual prompts.

**Reinforcement Learning from Human Feedback.** After web-scale pretraining, large language models are further enhanced through a two-step process: first, by supervised fine-tuning on demonstration data, and then by applying reinforcement learning to incorporate human feedback. Reinforcement learning from human feedback (RLHF) has proven to be an effective method for both improving the performance of large language models and aligning them with user preferences (Akrour et al., 2011; Christiano et al., 2017; Dubois et al., 2024; Dubey et al., 2024; Stiennon et al., 2020; Xiao et al., 2024b;a; Xu & Zhu, 2024). However, the alignment of text-to-image diffusion models with human preferences has been significantly less explored compared to LLMs. To bridge this gap, mutiple methods propose to apply supervised fine-tuning to improve text-to-image diffusion models. These approaches curate datasets by combining several methods, including preference models (Podell et al., 2023), pre-trained image models (Betker et al., 2023; Dong et al., 2023; Wu et al., 2023), such as image captioning models, and filtering data with the help of human experts (Dai et al., 2023). In the field of aligning and improving diffusion models, several studies have explored fine-tuning these models by leveraging reward models, either by directly increasing the reward of generated images (Clark et al., 2023; Prabhudesai et al., 2023; Hao et al., 2024) or through reinforcement learning techniques (Fan et al., 2024; Black et al., 2023). This process typically involves pretraining a reward model to capture specific human preferences. However, building a reliable reward model that accurately reflects human preferences is both challenging and computationally intensive. Furthermore, over-optimizing the reward model can result in severe issues, such as model collapse (Lee et al., 2023; Prabhudesai et al., 2023).

**Direct Preference Optimization.** Recently, several studies have proposed methods for directly optimizing preferences, such as Direct Preference Optimization (DPO) (Rafailov et al., 2024). These approaches bypass the need for a separate reward model training phase by directly fine-tuning models using preference data, often achieving better performance than RLHF-based methods (Ethayarajh et al., 2024; Azar et al., 2024; Zhao et al., 2023; Munos et al., 2023). Inspired by the success of these approaches, multiple recent methods directly adopt these preference learning methods originally designed for LLMs to fine-tune T2I diffusion models to align with human preferencess (Wallace et al., 2024; Yang et al., 2024; Li et al., 2024; Yuan et al., 2024; Gu et al., 2024). However, Moreover, directly adapting these algorithms from LLM domains results in an estimated loss. For instance, Diffusion-DPO is upper-bounded by the original DPO loss (Wallace et al., 2024). This estimation can lead to suboptimal performance when fine-tuning diffusion models to align with human preferences as shown in Figure 1 through empirical results. To address this, we propose Direct Score Preference Optimization (DSPO), a method for fine-tuning diffusion models by aligning their output distribution with human preferences using score matching. This approach is the first to apply preference learning from the perspective of score matching, offering a novel framework for designing effective preference learning algorithms for diffusion models.

## 3  NOTATIONS AND PRELIMINARIES

**Diffusion Model.** Denoising Diffusion Probabilistic Models (DDPMs) (Ho et al., 2020) represent the image generation process as a Markovian process. Starting with data $\mathbf{x}_0$, the forward process gradually adds noise using a predefined variance schedule, $\beta_1, \ldots, \beta_T$, which is defined as follows:

$$q\left(\mathbf{x}_{1:T} \mid \mathbf{x}_0\right) := \prod_{t=1}^{T} q\left(\mathbf{x}_t \mid \mathbf{x}_{t-1}\right), \quad q\left(\mathbf{x}_t \mid \mathbf{x}_{t-1}\right) := \mathcal{N}\left(\mathbf{x}_t; \sqrt{1-\beta_t}\mathbf{x}_{t-1}, \beta_t\mathbf{I}\right). \quad (1)$$

The training of diffusion models involves parameterizing the reverse process $p_\theta\left(\mathbf{x}_{t-1} \mid \mathbf{x}_t\right)$ using a neural network in DDPM (Ho et al., 2020), which is defined as:

$$p_\theta\left(\mathbf{x}_t \mid \mathbf{x}_{t+1}, \mathbf{c}\right) = \mathcal{N}\left(\mathbf{x}_t; \sqrt{\frac{\alpha_t}{\alpha_{t+1}}}\left(\mathbf{x}_{t+1} - \frac{\beta_{t+1}}{\sqrt{1-\bar{\alpha}_{t+1}}}\boldsymbol{\epsilon}_\theta\left(\mathbf{x}_{t+1}, \mathbf{c}, t+1\right)\right), \sigma_{t+1}^2\mathbf{I}\right), \quad (2)$$

where $\sigma_{t+1}^2 = \frac{1-\bar{\alpha}_t}{1-\bar{\alpha}_{t+1}}\beta_{t+1}$, $\alpha_t = 1 - \beta_t$, $\bar{\alpha}_t = \prod_{s=1}^{t}\alpha_s$. Then, the evidence lower bound (ELBO) is minimized to train the diffusion model with the following equation:

$$\mathcal{L}_{\text{DDPM}} = \mathbb{E}_{\mathbf{x}_0, t, \boldsymbol{\epsilon}}\left[\lambda(t)\left\|\boldsymbol{\epsilon} - \boldsymbol{\epsilon}_\theta\left(\mathbf{x}_t, t\right)\right\|^2\right], \quad (3)$$

where $\boldsymbol{\epsilon} \sim \mathcal{N}(0, \mathbf{I})$, $t \sim \mathcal{U}(0, T)$, $\mathbf{x}_t \sim q\left(\mathbf{x}_t \mid \mathbf{x}_0\right) = \mathcal{N}\left(\mathbf{x}_t; \sqrt{\bar{\alpha}_t}\mathbf{x}_0, (1-\bar{\alpha}_t)\mathbf{I}\right)$, $\lambda(t)$ is a time-dependent weighting function and $\theta$ represents learnable parameters.

**RLHF on T2I Diffusion Models.** RLHF typically involves fitting a reward model to human preference data and then fine-tuning the generative model to maximize expected reward through reinforcement learning. In the reward fitting process, human preferences can be modelled using the Bradley-Terry (BT) model (Bradley & Terry, 1952). To adapt the BT model to diffusion models, we define the posterior of human preferences for each time step $t$ with the following formula:

$$p_{\text{BT}}\left(\mathbf{x}_t^w \succ \mathbf{x}_t^l \mid \mathbf{c}\right) = \sigma\left(r\left(\mathbf{c}, \mathbf{x}_t^w\right) - r\left(\mathbf{c}, \mathbf{x}_t^l\right)\right), \quad (4)$$

where $\sigma(\cdot)$ denotes the sigmoid function, $\mathbf{c}$ is the textual prompt, $\mathbf{x}_t^w$ and $\mathbf{x}_t^l$ are a pair of winning and losing image samples at the time step $t$ of diffusion models.

After the reward function is learned, the generative model is optimized using reinforcement learning based on the reward feedback. By conceptualizing the denoising process of the diffusion model as a multi-step Markov Decision Process (MDP) and following Wallace et al. (2024); Fan et al. (2024); Yang et al. (2024); Li et al. (2024), we consider reward models at each step to define the objective:

$$\mathcal{L}_{\text{rlhf}} = \mathbb{E}_{\mathbf{c} \sim \mathcal{D}}\mathbb{E}_{p_\theta(\mathbf{x}_{0:T}|\mathbf{c})}\sum_{t=0}^{T-1} r\left(\mathbf{x}_t, \mathbf{c}\right) - \lambda\mathbb{D}_{\text{KL}}\left[p_\theta(\mathbf{x}_{0:T}|\mathbf{c})\|p_{\text{ref}}(\mathbf{x}_{0:T}|\mathbf{c})\right], \quad (5)$$

where $p_{\text{ref}}\left(\mathbf{x}_{0:T}\right)$ is the learnt distribution from pretrained diffusion models, $\mathbf{c}$ is the textual prompt sampled from dataset $\mathcal{D}$, $\mathbb{D}_{\text{KL}}\left[\cdot\|\cdot\right]$ represents the KL divergence between two distributions and $\lambda$

is the hyperparameter to control the weight of this KL term. We put more details about RLHF and modeling diffusion models as MDP into Appendix A due to space constraints.

**DPO on T2I Diffusion Models.** To simplify RLHF, DPO (Rafailov et al., 2024) uses the log-likelihood of the learning policy to implicitly represent the reward function. In the context of T2I diffusion models, the step-wise reward function for them can be defined as:

$$r(\mathbf{x}_t, \boldsymbol{c}) = \lambda \log \frac{p_\theta\left(\mathbf{x}_t \mid \mathbf{x}_{t+1}, \boldsymbol{c}\right)}{p_{\text{ref}}\left(\mathbf{x}_t \mid \mathbf{x}_{t+1}, \boldsymbol{c}\right)}. \tag{6}$$

Following this formulation, existing works (Wallace et al., 2024; Yang et al., 2024) adapt DPO algorithms, which aims to optimize $p_\theta$ based on the BT model in Equation (4), to diffusion models by framing them as MDPs. The objective is defined as follows:

$$\mathcal{L}_{\text{Diffusion}-\text{DPO}} = -\mathbb{E}\left[\log \sigma \left(\lambda \log \frac{p_\theta\left(\mathbf{x}_t^w \mid \mathbf{x}_{t+1}^w, \boldsymbol{c}\right)}{p_{\text{ref}}\left(\mathbf{x}_t^w \mid \mathbf{x}_{t+1}^w, \boldsymbol{c}\right)} - \lambda \log \frac{p_\theta\left(\mathbf{x}_t^l \mid \mathbf{x}_{t+1}^l, \boldsymbol{c}\right)}{p_{\text{ref}}\left(\mathbf{x}_t^l \mid \mathbf{x}_{t+1}^l, \boldsymbol{c}\right)}\right)\right], \tag{7}$$

where $\left(\mathbf{x}_0^w, \mathbf{x}_0^l, \boldsymbol{c}\right) \sim \mathcal{D}$, $t \sim \mathcal{U}(0, T)$, $\mathbf{x}_{t,t+1}^w \sim p\left(\mathbf{x}_{t,t+1}^w \mid \mathbf{x}_0^w\right)$ and $\mathbf{x}_{t-1,t}^l \sim p\left(\mathbf{x}_{t,t+1}^l \mid \mathbf{x}_0^l\right)$. To simplify notation, we use $\mathbf{x}_t$ to represent $\mathbf{x}_t^w$ in the following section.

## 4 METHOD

We introduce Direct Score Preference Optimization (DSPO), a preference learning algorithm grounded in score matching principles, tailored for fine-tuning diffusion models. Our approach begins by defining a target human-preferred score function, which combines the ground-truth data distribution score with a preference model. We then fine-tune the pretrained score models to align with this target, guiding the diffusion process toward generating human-preferred outputs. The illustration of the model framework is displayed in Figure 3.

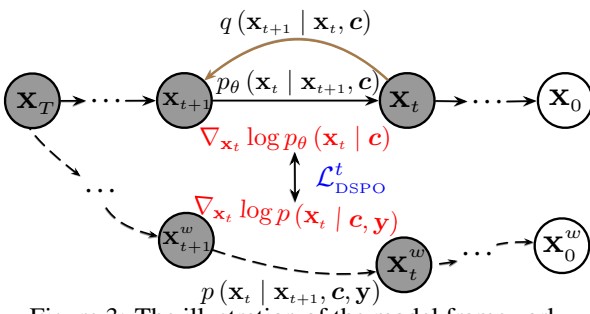

Figure 3: The illustration of the model framework.

### 4.1 HUMAN PREFERENCE SCORE MODEL

In this section, we introduce the target human preference score model, which is aligned with human preferences. Our goal is to use this target model for fine-tuning the pretrained score or diffusion models to match the this target model. Before presenting this, we first introduce the score model used to leverage the connection between diffusion models and score matching (Song et al., 2020). And the corresponding score function can be derived as $\nabla_{\mathbf{x}_t} \log p_\theta\left(\mathbf{x}_t \mid \boldsymbol{c}\right)$. By incorporating conditional constraints in T2I diffusion models, we can get the score model $\nabla_{\mathbf{x}_t} \log p_\theta\left(\mathbf{x}_t \mid \boldsymbol{c}, \mathbf{y}\right)$ for the conditional variables $\mathbf{y}$. This can be derived using Bayes' rule as follows:

$$\nabla_{\mathbf{x}_t} \log p_\theta\left(\mathbf{x}_t \mid \boldsymbol{c}, \mathbf{y}\right) = \nabla_{\mathbf{x}_t} \log p_\theta\left(\mathbf{x}_t \mid \boldsymbol{c}\right) + \nabla_{\mathbf{x}_t} \log p\left(\mathbf{y} \mid \mathbf{x}_t, \boldsymbol{c}\right). \tag{8}$$

Based on this formulation, we can treat human-preferred properties as constrained conditions for T2I diffusion models, which is represented by the variable $\mathbf{y}$. The probability of whether the input images $p\left(\mathbf{y} \mid \mathbf{x}_t, \boldsymbol{c}\right)$ align with human preferences can be obtained using Equation (4):

$$p\left(\mathbf{y} \mid \mathbf{x}_t, \boldsymbol{c}\right) = p\left(\mathbf{y} \mid \mathbf{x}_t, \mathbf{x}_t^l, \boldsymbol{c}\right) = p\left(\mathbf{x}_t \succ \mathbf{x}_t^l \mid \mathbf{x}_t^l, \boldsymbol{c}\right) = \sigma\left(r\left(\boldsymbol{c}, \mathbf{x}_t\right) - r\left(\boldsymbol{c}, \mathbf{x}_t^l\right)\right). \tag{9}$$

By treating the variable $\mathbf{y}$ as human-preferred conditions, we can derive a human preference score model. To achieve image generation based on human-preferred conditions in a training-free manner, we can first naively train a preference model to estimate $p_\phi\left(\mathbf{x}_t \succ \mathbf{x}_t^l \mid \mathbf{x}_t^l, \boldsymbol{c}\right)$. This trained model can then be used to replace the second term in Equation (8), following the widely-used classifier guidance method for diffusion models (Dhariwal & Nichol, 2021). However, this approach has two major drawbacks: (i) To determine the probability of human-preferred images, $p_\phi$, we must input $\mathbf{x}_t^l$ at each time step, which requires providing negative samples for every target prompt—a task that is impractical in real-world applications. (ii) Calculating the gradient of the trained classifier increases inference time, and training a robust classifier for all reverse steps, especially for highly noisy inputs at the initial steps, is a significant challenge (Ho & Salimans, 2022).

## 4.2 DIRECT SCORE PREFERENCE OPTIMIZATION

In this paper, we focus on a novel fine-tuning method instead of training-free method for pretrained T2I diffusion models to better align with human preferences. Our approach ensures that the fine-tuning objective is consistent with the objective used during the pretraining stage, unlike current methods that adapt fine-tuning techniques from LLMs (Wallace et al., 2024; Yang et al., 2024; Li et al., 2024), which differ from the pretraining objectives and may lead to suboptimal results. Specifically, we propose fine-tuning the pretrained T2I diffusion model to match the target human preference score model for each time step $t$ introduced in Section 4.1, which is defined as follows:

$$\min_{\theta} \omega(t) \|\nabla_{\mathbf{x}_t} \log p_{\theta}(\mathbf{x}_t \mid \boldsymbol{c}) - (\nabla_{\mathbf{x}_t} \log p(\mathbf{x}_t \mid \boldsymbol{c}) + \gamma \nabla_{\mathbf{x}_t} \log p(\mathbf{y} \mid \mathbf{x}_t, \boldsymbol{c}))\|_2^2, \quad (10)$$

where $\omega(t)$ is a time-dependent function for score matching as introduced in Song et al. (2020). $\gamma$ is used to control the weight of conditional constraints towards human preferred image generation and $p(\mathbf{y} \mid \mathbf{x}_t, \boldsymbol{c}) = \sigma\left(r\left(\boldsymbol{c}, \mathbf{x}_t\right) - r\left(\boldsymbol{c}, \mathbf{x}_t^l\right)\right)$ is represented as Equation (9) for human preference conditions. To avoid training an extra probability model, we use the implicit reward $r(\boldsymbol{c}, \mathbf{x}_t)$ defined in Equation (6) to replace the reward model in Equation (9). Based on the reverse process $p_{\theta}(\mathbf{x}_t \mid \mathbf{x}_{t+1}, \boldsymbol{c})$ of T2I diffusion models in Equation (2), we can get the following reward $r(\mathbf{x}_t, \boldsymbol{c})$:

$$r(\mathbf{x}_t, \boldsymbol{c}) = -\frac{\lambda \beta_{t+1}}{2(1 - \bar{\alpha}_t)} \frac{\alpha_t}{\alpha_{t+1}} \left(\|\boldsymbol{\epsilon}_{\theta}(\boldsymbol{x}_{t+1}, t+1) - \boldsymbol{\epsilon}_{t+1}\|_2^2 - \|\boldsymbol{\epsilon}_{\text{ref}}(\boldsymbol{x}_{t+1}, t+1) - \boldsymbol{\epsilon}_{t+1}\|_2^2\right) \quad (11)$$

Details about achieving this equation are put into Appendix B.1. We use the score function of the true data distribution instead of the pretrained model $p_{\text{ref}}$ in the second term of Equation (10) because the pretrained model may not accurately reflect the true data distribution's score function. Based on Equation (11), we get the following objective after derivations on Equation (10):

$$\min_{\theta} A(t) \left\|B(t)\left(\boldsymbol{\epsilon}_{\theta,t+1} - \boldsymbol{\epsilon}_{t+1}\right) - \lambda\gamma\left(1 - \sigma\left(r\left(\boldsymbol{c}, \mathbf{x}_t\right) - r\left(\boldsymbol{c}, \mathbf{x}_t^l\right)\right)\left(\boldsymbol{\epsilon}_{\theta,t+1} - \boldsymbol{\epsilon}_{\text{ref},t+1}\right)\right)\right\|_2^2, \quad (12)$$

where $A(t) = \omega(t)\frac{1}{4\sigma_{t+1}^4}\frac{\alpha_t}{\alpha_{t+1}}\frac{\beta_{t+1}^2}{1 - \bar{\alpha}_{t+1}}$, $\boldsymbol{\epsilon}_{\theta,t+1} = \boldsymbol{\epsilon}_{\theta}(\mathbf{x}_{t+1}, \boldsymbol{c}, t+1)$, $\lambda$ is a hyperparameter that determines the weight used to control the KL divergence in Equation (5), similarly for $\boldsymbol{\epsilon}_{\text{ref},t+1}$ and $r(\cdot)$ is defined in Equation (11). $B(t)$ is a time-dependent parameter whose specific form is provided in Appendix B.2 due to space constrains. Based on our empirical findings and to further simplify the loss function, we omit $B(t)$ in our experiment, arriving at our final objective:

$$\mathcal{L}_{\text{DSPO}}^t = A(t) \left\|\boldsymbol{\epsilon}_{\theta,t+1} - \boldsymbol{\epsilon}_{t+1} - \lambda\gamma\left(1 - \sigma\left(r\left(\boldsymbol{c}, \mathbf{x}_t\right) - r\left(\boldsymbol{c}, \mathbf{x}_t^l\right)\right)\left(\boldsymbol{\epsilon}_{\theta,t+1} - \boldsymbol{\epsilon}_{\text{ref},t+1}\right)\right)\right\|_2^2, \quad (13)$$

We set $\gamma = 1$ to avoid extra hyperparameter for fine-tuning diffusion models. The derivations are put into Appendix B.2. Following DDPM (Ho et al., 2020) and Diffusion-DPO (Wallace et al., 2024), we disregard $A(t)$ and the associated parameters about $\alpha_t$ and $\beta_t$ at the beginning of Equation (13). In our settings where only preference data are accessible, we have our following final objectives:

$$\min_{\theta} \mathbb{E}_{(\mathbf{x}_0^w, \mathbf{x}_0^l, \boldsymbol{c}) \sim \mathcal{D}, t \sim \mathcal{U}(0,T), \mathbf{x}_t \sim p(\mathbf{x}_t | \mathbf{x}_0^w, \boldsymbol{c}), \mathbf{x}_t^l \sim p(\mathbf{x}_t^l | \mathbf{x}_0^l, \boldsymbol{c})} \mathcal{L}_{\text{DSPO}}^t. \quad (14)$$

## 4.3 THEORETICAL ANALYSIS

In this section, we provide a theoretical analysis about DSPO. Our analyses show the relation with RLHF objectives on diffusion models in Equation (5). Specifically, We demonstrate that, under certain conditions, minimizing DSPO by sampling data from the trained diffusion model and distilling the score from the reference model shares similar optimization directions with maximizing RLHF objectives in Equation (5). Because of space constraints, all proofs are put into the Appendix C.

Next, we start by deriving an equivalent form of the RLHF objective on T2I diffusion models in Equation (5) by rearranging the elements in this equation. We can view the RLHF objective as optimizing a reverse KL-divergence between $p_{\theta}(\cdot)$ and $p^*(\cdot)$ from the probability matching perspective:

$$\mathcal{L}_{\text{rlhf}} = \mathbb{E}_{\boldsymbol{c} \sim \mathcal{D}} \mathbb{E}_{p_{\theta}(\mathbf{x}_{0:T} | \boldsymbol{c})} \sum_{t=0}^{T-1} -\lambda \mathbb{D}_{\text{KL}}\left[p_{\theta}(\mathbf{x}_t \mid \mathbf{x}_{t+1}, \boldsymbol{c}) \| p^*(\mathbf{x}_t \mid \mathbf{x}_{t+1}, \boldsymbol{c})\right] + \log Z(\boldsymbol{c}), \quad (15)$$

where $Z(\boldsymbol{c}) = \int \exp(\sum_{t=0}^{T-1} r(\mathbf{x}_t, \boldsymbol{c})/\lambda) p_{\text{ref}}(\mathbf{x}_{0:T} | \boldsymbol{c}) \mathrm{d}\mathbf{x}_{0,T}$ is independent of learnable parameter $\theta$ and $p^*(\mathbf{x}_t \mid \mathbf{x}_{t+1}, \boldsymbol{c}) \propto p_{\text{ref}}(\boldsymbol{x}_t \mid \boldsymbol{x}_{t+1}, \boldsymbol{c}) e^{(r(\boldsymbol{x}_t, \boldsymbol{c}))/\lambda}$. The details of derivations for this equation

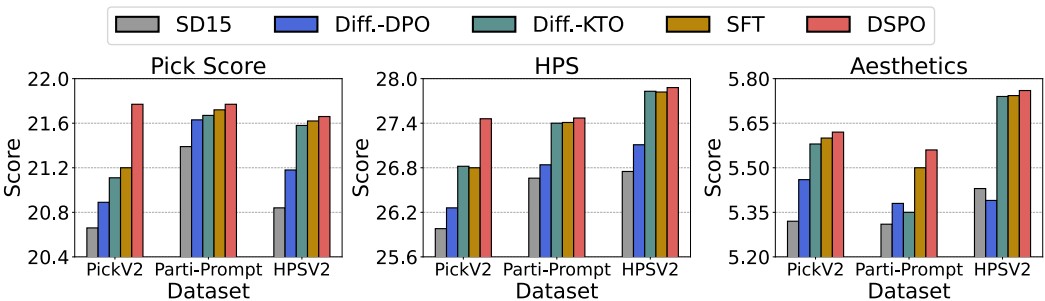

Figure 4: Reward score comparisons on all datasets for various baselines by different reward models.

are put into Appendix C.1. In the following theoretical demonstration, we show that optimizing our DSPO shares the same optimization direction as maximizing the RLHF objective. This is achieved by matching the hyperparameters $\omega(t)$ and $\gamma$ with those in the RLHF framework.

**Theorem 1** *Following $\omega(t) = 2\sigma_{t+1}^2/\lambda$, $\gamma = 1/2\lambda$, reward model $r(\cdot)$ as defined in Equation (4) and $p_{\text{data}}(\cdot)$ as the reference model for RLHF of T2I diffusion models in Equation (15), the gradient of DSPO objective in Equation (13) by sampling data from $p_\theta$ satisfies:*

$$\nabla_\theta \mathcal{L}_{\text{rlhf}} = \nabla_\theta \mathbb{E}_{\boldsymbol{c}\sim\mathcal{D}} \mathbb{E}_{p_\theta(\mathbf{x}_{0:T}|\boldsymbol{c})} \sum_{t=0}^{T-1} -\mathcal{L}_{\text{DSPO}}^t. \tag{16}$$

Theorem 1 indicates the optimization direction for $p_\theta$ during intermediate steps of minimizing $\mathcal{L}_{\text{DSPO}}$, when sampling data from $p_\theta$, aligns with the direction required to maximize $\mathcal{L}_{\text{rlhf}}$ asymptotically, given a sufficiently large dataset. Our final optimized loss in Equation (14) is an empirical estimate of the loss in Equation (16) on preference feedback by sampling pairs of images, $\mathbf{x}^w$, $\mathbf{x}^l$. Moreover, we consider the reward model $r(\cdot) = p_{\text{BT}}\left(\mathbf{x}_t^w \succ \mathbf{x}_t^l \mid \boldsymbol{c}\right)$ as defined in Equation (4), which evaluates a pair of images based on human feedback. Alternatively, we can retain the original reward model format, i.e. $r(\mathbf{x})$, which assesses individual images rather than comparing pairs based on human feedback. Maximizing the RLHF objective while maintaining the original reward structure is equivalent to minimizing our objective by setting $p(\mathbf{y} \mid \mathbf{x}_t, \boldsymbol{c}) = \exp\left(r(\mathbf{x}_t, \boldsymbol{c})/\lambda\right)/Z(\boldsymbol{c})$, where $Z(\boldsymbol{c}) = \int \exp\left(r(\mathbf{x}_t, \boldsymbol{c})/\lambda\right) d\mathbf{x}_t$, under the same condition in Theorem 1. Detailed derivation are put into Appendix C.3. We conduct an ablation study on this approach in Section 5.3.

## 5 EXPERIMENT

### 5.1 EXPERIMENTAL SETUP

**Datasets and Models.** We fine-tune Stable Diffusion v1.5 (SD1.5) using the DSPO objective on image pairs based on human feedback, as described in Equation (14), following the Diffusion-DPO approach (Wallace et al., 2024). This is done using the Pick-a-Pic v2 (Pick V2) dataset (Kirstain et al., 2023), which contains image preference pairs for each prompt. To evaluate the model, we use test prompts from Pick V2, the HPSV2 benchmark prompts (Wu et al., 2023), and the Parti-Prompts dataset (Yu et al., 2022). We conduct the image editing experiment with text instructions on InstructPix2Pix dataset (Brooks et al., 2023). The details of dataset are put into Appendix D.2.

**Baselines.** We evaluate the effectiveness of aligning T2I diffusion models with DSPO by comparing the generations from our DSPO aligned model to those from other existing methods, including the original pretrained SD1.5 or SDXL, supervised fine-tuning (SFT) approaches, Diffusion-DPO (Wallace et al., 2024), MaPO (Hong et al., 2024) and Diffusion-KTO (Li et al., 2024). Note that when training the SDXL model, the quality of training data in PickV2 is lower than that of images generated by SDXL. Therefore, we use the reference model for $p(\mathbf{x}_t|\mathbf{c})$. A detailed discussion of the training method on SDXL is provided in Appendix D.1.

**Evaluation.** To assess human preference alignment, we perform Text-to-image (T2I) generation and text-guided image editing. We evaluated each task with several metrics, including Pick Score (Kirstain et al., 2023), HPSV2 (Wu et al., 2023), LAION Aesthetics Score (Schuhmann, 2022), CLIP (Radford et al., 2021), and ImageReward (Xu et al., 2024). For each reward model, we

Table 1: Win-rate comparison between DSPO and other baselines versus SD1.5, evaluated on different reward models using prompts from the PickV2, HPSV2, and Parti-Prompt datasets (T2I Generation). For simplicity, "Diff." represents "Diffusion". Best results are highlighted in **boldface**.

| Dataset | Method | Pick Score | HPS | Aesthetics | CLIP | Image Reward |
|---------|--------|-----------|-----|------------|------|--------------|
| PickV2 | SFT | 70.20 | 84.20 | 75.80 | 61.20 | 76.40 |
| | Diff.-DPO | 71.60 | 70.20 | 66.20 | 58.80 | 63.60 |
| | Diff.-KTO | 71.40 | 84.40 | 72.60 | 60.02 | 77.00 |
| | DSPO | **73.60** | **84.80** | **76.20** | **61.80** | **78.00** |
| Parti-Prompt | SFT | 64.27 | 85.72 | 75.74 | 54.72 | 71.38 |
| | Diff.-DPO | 61.18 | 66.48 | 60.42 | **55.45** | 62.19 |
| | Diff.-KTO | 64.80 | 86.16 | 72.92 | 54.34 | 71.51 |
| | DSPO | **65.32** | **87.50** | **76.96** | 54.86 | **71.75** |
| HPSV2 | SFT | 79.03 | 91.97 | 78.56 | 60.47 | 80.78 |
| | Diff.-DPO | 76.06 | 72.13 | 66.00 | 58.50 | 64.22 |
| | Diff.-KTO | 79.18 | 92.15 | 77.87 | 59.28 | 81.96 |
| | DSPO | **79.90** | **92.56** | **80.59** | **61.13** | **82.31** |

Table 2: Win-rate comparison between DSPO and other baselines versus SDXL, evaluated on different reward models using prompts from the PickV2, HPSV2, and Parti-Prompt datasets (T2I Generation). **Note that the DSPO in this table is fine-tuned on SDXL**.

| Dataset | Method | Pick Score | HPS | Aesthetics | CLIP | Image Reward |
|---------|--------|-----------|-----|------------|------|--------------|
| PickV2 | SFT | 20.80 | 40.60 | 23.20 | 44.80 | 34.40 |
| | Diff.-DPO | **75.20** | 76.20 | 54.10 | 59.40 | 65.20 |
| | MaPO | 54.40 | 69.60 | **68.20** | 51.20 | 61.40 |
| | DSPO | 74.00 | **80.00** | 54.20 | **59.60** | **68.60** |
| Parti-Prompt | SFT | 17.03 | 33.02 | 27.81 | 36.58 | 37.18 |
| | Diff.-DPO | 65.44 | 74.08 | 56.86 | **60.54** | 66.85 |
| | MaPO | 58.34 | 66.54 | **68.23** | 47.43 | 58.64 |
| | DSPO | **67.46** | **81.80** | 57.84 | 55.02 | **73.47** |
| HPSV2 | SFT | 18.18 | 45.28 | 26.72 | 39.13 | 47.22 |
| | Diff.-DPO | 70.31 | 80.81 | 50.78 | **59.31** | 68.75 |
| | MaPO | 59.62 | 77.90 | **62.31** | 50.90 | 62.09 |
| | DSPO | **72.59** | **83.47** | 51.41 | 57.34 | **70.09** |

report both the average scores for all models and win rates between DSPO or baselines and Stable Diffusion v1.5. For a fair comparison, we use the default hyperparameters for the T2I diffusion model to sample images across all baselines and DSPO as used in Rafailov et al. (2024), ensuring consistency in evaluation, i.e., guidance scale as 7.5 and the number of sampling steps as 50. Note that for our evaluation experiments, we directly use the checkpoints for Diffusion-DPO and Diffusion-KTO provided by the authors. Additionally, we train the SFT model following Diffusion-DPO for our evaluations. We conduct five sampling runs for each algorithm using different seeds, and the average results are reported. Implementation details of DSPO are provided in Appendix D.3.

## 5.2 PERFORMANCE COMPARISON ON HUMAN PREFERENCE ALIGNMENT

We present the results of real rewards from various reward models across all datasets of T2I generation in the Figure 4, comparing DSPO with SFT, Diff.-DPO, Diff.-KTO, and SD1.5. Due to space constraints, additional reward score results (CLIP and Image Reward) are provided in Appendix E.2. The results consistently show that DSPO outperforms all baselines. Notably, our fine-tuned T2I diffusion model significantly surpasses the original base model SD1.5. For example, SD1.5 achieves an

Table 3: Computational costs of Diffusion-DPO and DSPO using 1 NVIDIA A100s. Training time ("Time") for each optimization step and peak GPU memory without the model ("GPU Mem.") measured with 16 batch size and 128 accumulation gradient step in fine-tuning SD15 on PickV2.

| | Diffusion-DPO | DSPO |
|---|---|---|
| **Time** (↓) | 4.15 min | 4.18 min |
| **GPU Mem.** (↓) | 60.2 | 60.5 |

Table 4: Win-rate comparison of InstructPix2Pix dataset for text-guided image editing.

| Dataset | Method | Pick Score | HPS | Aesthetics | CLIP | Image Reward |
|---------|--------|-----------|------|-----------|------|-------------|
| InstructPix2Pix | SFT | 57.10 | 66.60 | 73.10 | 48.60 | **61.10** |
| | Diff.-DPO | 51.40 | 52.00 | 52.80 | 46.80 | 47.00 |
| | Diff.-KTO | 53.60 | 69.20 | 72.20 | 50.00 | 61.00 |
| | DSPO | **58.40** | **69.30** | **73.80** | **51.30** | **61.10** |

Table 5: Win-rate comparison between DSPO and its variant DSPO-E versus SD1.5, evaluated across different reward models using prompts from the PickV2, HPSV2, and Parti-Prompt datasets.

| Dataset | Method | Pick Score | HPS | Aesthetics | CLIP | Image Reward |
|---------|--------|-----------|------|-----------|------|-------------|
| PickV2 | DSPO-E | 70.20 | 84.00 | 73.20 | 60.60 | 75.80 |
| | DSPO | **73.60** | **84.80** | **76.20** | **61.80** | **78.00** |
| Parti-Prompt | DSPO-E | 62.86 | 85.31 | 75.91 | 54.81 | 71.69 |
| | DSPO | **65.32** | **87.50** | **76.96** | **54.86** | **71.75** |
| HPSV2 | DSPO-E | 75.06 | 91.28 | 77.65 | 59.59 | 80.93 |
| | DSPO | **79.90** | **92.56** | **80.59** | **61.13** | **82.31** |

Image Reward score of only 0.018, while DSPO attains a much higher score of 0.568. Additionally, DSPO outperforms both Diff.-DPO and Diff.-KTO, which also use preference learning algorithms. This validates the effectiveness of our model in aligning with human preferences.

Table 1 presents the win-rate comparison of SFT, Diffusion-DPO (Diff.-DPO), Diffusion-KTO (Diff.-KTO), and DSPO aligned SD1.5 against the original pretrained SD1.5 for T2I generation. In general, DSPO achieves the best performance compared to recent baselines across all datasets and nearly all reward models, demonstrating its effectiveness. Notably, DSPO significantly enhances alignment for the base SD1.5 model, achieving win-rates as high as 92.56% according to the HPSV2 reward model. Furthermore, DSPO outperforms existing baselines, such as Diff.-DPO and Diff.-KTO, which adapt algorithms from LLM domains to diffusion models, across nearly all reward models. Specifically, DSPO achieves an absolute win-rate improvement of 16.54% and 4.04% over Diff.-DPO and Diff.-KTO, respectively, on the Parti-Prompt dataset for the Aesthetics reward model. This validates the motivation that matching the loss objectives during both the pretraining and fine-tuning stages of T2I diffusion models enhances overall model performance. Table 2 presents the results of DSPO fine-tuned on SDXL, alongside the corresponding baselines. The results demonstrate that our models outperform the baselines across most metrics on all three datasets, further validating the effectiveness of our approach. We also conduct memory and wall-clock experiments in Table 3. Compared to Diffusion-DPO, DSPO shows comparable runtime and memory usage.

Table 4 presents the win-rate comparison results for the text-guided image editing task. Similarly, DSPO outperforms all baselines across different reward models, further demonstrating its effectiveness and potential applicability to a wide range of text-based image generation tasks.

## 5.3 ABLATION STUDY

We display the results of DSPO and its variant DSPO-E performance in Table 5. Specifically, as outlined in Section 4.3, we can express $p\left(\mathbf{y} \mid \mathbf{x}_t, \boldsymbol{c}\right)$ as an energy-based distribution $p\left(\mathbf{y} \mid \mathbf{x}_t, \boldsymbol{c}\right) = \exp\left(r(\mathbf{x}_t, \boldsymbol{c})/\lambda\right)/Z(\boldsymbol{c})$, where $Z(\boldsymbol{c}) = \int \exp\left(r(\mathbf{x}_t, \boldsymbol{c})/\lambda\right) d\mathbf{x}_t$. Optimizing this variant of DSPO follows the same optimization direction as the RLHF objective, as demonstrated in Theorem 1 without assuming reward functions as BT models. Further details on this variant are provided in Appendix C.3 and we denote it as DSPO-E. We observe that our models outperform the DSPO-E variant on all datasets for all reward models, highlighting the effectiveness of using the BT model for human preference learning, as outlined in Equation (4). Unlike DSPO-E, which relies on implicit reward models for single images, DSPO leverages image pairs from human preference feedback, providing richer information and enhancing overall performance.

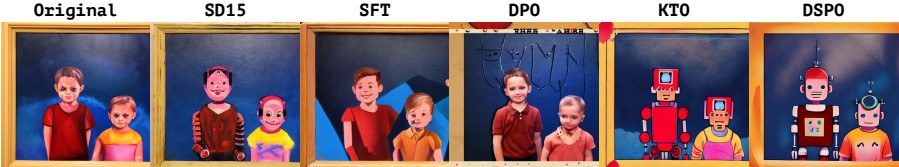

Figure 5: We show the images generated by different models from one prompt, which is "Frontal portrait of an anime girl with pink hair and sunglasses wearing a white tshirt.".

Figure 6: We show the text-guided image editing task with the prompt "The siblings are all robots".

## 5.4 Qualitative Analysis

Figure 5 showcase the qualitative performance of our model on T2I image generation used in this paper. Compared to the baseline methods, DSPO exhibits a clear enhancement in image quality, which is even more pronounced than the improvements reflected in the reward scores. Specifically, DSPO accurately generates details such as sunglasses, pink hair, and an anime girl, while simultaneously creating a more visually appealing image compared to other baselines. Furthermore, Figure 6 presents the qualitative results of DSPO on text-guided image editing. In comparison to other baselines, DSPO not only faithfully adheres to the textual description when transforming siblings into robots, but also generates more realistic and visually acceptable images. In summary, the advantages of generated images from DSPO are particularly evident in key aspects such as alignment, visual appeal, and the intricacy of details within each image. These qualitative results emphasize DSPO's ability to generate images that are not only contextually accurate but also visually superior to those produced by existing models. Additional prompts and qualitative results for both of experiments are provided in Appendix E.3 due to space constrains in the main paper.

## 6 Conclusion

In this paper, we propose Direct Score Preference Optimization (DSPO), a novel approach for fine-tuning diffusion-based text-to-image (T2I) models by aligning their pretraining and fine-tuning objectives through score matching. By leveraging the inherent score function of diffusion models and incorporating human preference feedback without relying on complex reward models, DSPO addresses the performance gaps observed with existing fine-tuning techniques such as Diffusion-DPO. We theoretically demonstrate that optimizing DSPO shares the same optimization directions as optimizing Reinforcement Learning from Human Feedback (RLHF) objectives, ensuring the effectiveness of the fine-tuning process. Our empirical results show that DSPO consistently outperforms other preference learning methods, confirming its capability to enhance image generation for human preferences. This approach offers a new direction for preference alignment in diffusion models, bridging the gap between pretraining and fine-tuning for more user-aligned outputs.

## Acknowledgment

The work of Vasant G Honavar, Huaisheng Zhu and Teng Xiao was supported in part by grants from the National Science Foundation (2226025, 2225824), the National Center for Advancing Translational Sciences, and the National Institutes of Health (UL1 TR002014).

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

## A    OMITTED DETAILS OF RLHF ON DIFFUSION MODELS

In this section, we follow the approach of Black et al. (2023); Fan et al. (2024) to model the diffusion reverse process under the conditional generation setting as a Markov Decision Process (MDP), defined by $\mathcal{M} = (\mathbb{S}, \mathbb{A}, \mathcal{P}, r, \rho)$. Specifically, $\pi$ represents the policy network and the diffusion reverse chain is $\{x_t\}_{t=T}^0$ with length $T$. This MDP can be defined as:

$$
\begin{aligned}
\mathbf{s}_t &\triangleq (\boldsymbol{c}, t, \mathbf{x}_t) & \pi\left(\mathbf{a}_t \mid \mathbf{s}_t\right) &\triangleq p_\theta\left(\mathbf{x}_{t-1} \mid \mathbf{x}_t, \boldsymbol{c}\right) & \mathcal{P}\left(\mathbf{s}_{t+1} \mid \mathbf{s}_t, \mathbf{a}_t\right) &\triangleq \left(\delta_{\boldsymbol{c}}, \delta_{t-1}, \delta_{\mathbf{x}_{t-1}}\right) \\
\mathbf{a}_t &\triangleq \mathbf{x}_{t-1} & \rho_0\left(\mathbf{s}_0\right) &\triangleq (p(\boldsymbol{c}), \delta_T, \mathcal{N}(\mathbf{0}, \mathbf{I})) & R\left(\mathbf{s}_t, \mathbf{a}_t\right) &\triangleq r(\mathbf{x}_{t-1}, \boldsymbol{c}),
\end{aligned}
\tag{17}
$$

where $\delta(\cdot)$ is the measure delta measure and $\mathcal{P}\left(\mathbf{s}_{t+1} \mid \mathbf{s}_t, \mathbf{a}_t\right)$ is a deterministic transition. Different from previous works, we represent each step's reward model as $r(\mathbf{x}_{t-1}, \boldsymbol{c})$. Based on this MDP, we get the objective for RLHF diffusion models following Fan et al. (2024):

$$
\mathbb{E}_{\boldsymbol{c} \sim \mathcal{D}} \mathbb{E}_{p_\theta(\mathbf{x}_{0:T}|\boldsymbol{c})} \sum_{t=1}^T r\left(\mathbf{x}_{t-1}, \mathbf{c}\right) - \lambda \mathbb{D}_{\mathrm{KL}}\left[p_\theta(\mathbf{x}_{0:T}|\boldsymbol{c}) \| p_{\mathrm{ref}}(\mathbf{x}_{0:T}|\boldsymbol{c})\right].
\tag{18}
$$

For notation simplicity in the following part, we set the range of $t$ from 0 to $T-1$ and get the following equation:

$$
\mathbb{E}_{\boldsymbol{c} \sim \mathcal{D}} \mathbb{E}_{p_\theta(\mathbf{x}_{0:T}|\boldsymbol{c})} \sum_{t=0}^{T-1} r\left(\mathbf{x}_t, \mathbf{c}\right) - \lambda \mathbb{D}_{\mathrm{KL}}\left[p_\theta(\mathbf{x}_{0:T}|\boldsymbol{c}) \| p_{\mathrm{ref}}(\mathbf{x}_{0:T}|\boldsymbol{c})\right]
\tag{19}
$$

## B    DERIVATIONS IN SECTION 4.2

### B.1    DERIVATIONS OF EQUATION (11)

In this section, we provide a detailed derivation of Equation (10) for the implicit reward model:

$$
\mathbb{E}_{\boldsymbol{c} \sim \mathcal{D}} \mathbb{E}_{p_\theta(\mathbf{x}_{0:T}|\boldsymbol{c})} \sum_{t=0}^{T-1} r\left(\mathbf{x}_t, \mathbf{c}\right) - \lambda \mathbb{D}_{\mathrm{KL}}\left[p_\theta(\mathbf{x}_{0:T}|\boldsymbol{c}) \| p_{\mathrm{ref}}(\mathbf{x}_{0:T}|\boldsymbol{c})\right].
\tag{20}
$$

Following DDPM (Ho et al., 2020), the definition of $p_\theta\left(\mathbf{x}_t \mid \mathbf{x}_{t+1}, \boldsymbol{c}\right)$ is as follows:

$$
\begin{aligned}
p_\theta\left(\mathbf{x}_t \mid \mathbf{x}_{t+1}, \boldsymbol{c}\right) &= \mathcal{N}\left(\mathbf{x}_t; \sqrt{\frac{\alpha_t}{\alpha_{t+1}}}\left(\mathbf{x}_{t+1} - \frac{\beta_{t+1}}{\sqrt{1 - \bar{\alpha}_{t+1}}} \boldsymbol{\epsilon}_\theta\left(\mathbf{x}_{t+1}, \boldsymbol{c}, t+1\right)\right), \sigma_{t+1}^2 \mathbf{I}\right) \\
&= \frac{1}{\left(\sqrt{2\pi\sigma_{t+1}^2}\right)^d} \exp\left(-\frac{1}{2\sigma_{t+1}^2}\left\|\mathbf{x}_t - \sqrt{\frac{\alpha_t}{\alpha_{t+1}}}\left(\mathbf{x}_{t+1} - \frac{\beta_{t+1}}{\sqrt{1 - \bar{\alpha}_{t+1}}} \boldsymbol{\epsilon}_\theta\left(\mathbf{x}_{t+1}, \boldsymbol{c}, t+1\right)\right)\right\|_2^2\right)
\end{aligned}
\tag{21}
$$

where $\sigma_{t+1}^2 = \frac{1-\bar{\alpha}_t}{1-\bar{\alpha}_{t+1}}\beta_{t+1}$ and $d$ is the dimension of the image.

We approximate $\mathbf{x}_t$ with its posterior mean $\mathbb{E}\left[\mathbf{x}_t \mid \mathbf{x}_{t+1}, \mathbf{x}_0\right] = \sqrt{\frac{\alpha_t}{\alpha_{t+1}}}\left(\boldsymbol{x}_{t+1} - \frac{\beta_{t+1}}{\sqrt{1-\bar{\alpha}_{t+1}}}\boldsymbol{\epsilon}_{t+1}\right)$. Then, we can get the estimated $p_\theta\left(\mathbf{x}_t \mid \mathbf{x}_{t+1}, \boldsymbol{c}\right)$ as:

$$
\frac{\lambda}{\left(\sqrt{2\pi\sigma_{t+1}^2}\right)^d} \exp\left(-\frac{1}{2}\frac{\beta_{t+1}}{(1-\bar{\alpha}_t)}\frac{\alpha_t}{\alpha_{t+1}}\left\|\boldsymbol{\epsilon}_\theta\left(\boldsymbol{x}_{t+1}, t+1\right) - \boldsymbol{\epsilon}_{t+1}\right\|_2^2\right).
\tag{22}
$$

Therefore, the reward function $r(\cdot) = \lambda\left(\log p_\theta\left(\mathbf{x}_t \mid \mathbf{x}_{t+1}, \boldsymbol{c}\right) - \log p_{\mathrm{ref}}\left(\mathbf{x}_t \mid \mathbf{x}_{t+1}, \boldsymbol{c}\right)\right)$ can be represented as by using the estimated $p_\theta\left(\mathbf{x}_t \mid \mathbf{x}_{t+1}, \boldsymbol{c}\right)$:

$$
r(\mathbf{x}_t, \boldsymbol{c}) = -\frac{\lambda}{2}\frac{\beta_{t+1}}{(1-\bar{\alpha}_t)}\frac{\alpha_t}{\alpha_{t+1}}\left(\left\|\boldsymbol{\epsilon}_\theta\left(\boldsymbol{x}_{t+1}, t+1\right) - \boldsymbol{\epsilon}_{t+1}\right\|_2^2 - \left\|\boldsymbol{\epsilon}_{\mathrm{ref}}\left(\boldsymbol{x}_{t+1}, t+1\right) - \boldsymbol{\epsilon}_{t+1}\right\|_2^2\right)
\tag{23}
$$

### B.2    DERIVATIONS OF EQUATION (12)

In this section, we provide a detailed derivation of Equation (12) for our loss:

$$
\begin{aligned}
&\min_\theta \omega(t)\left\|\nabla_{\mathbf{x}_t} \log \frac{p_\theta(\mathbf{x}_t \mid \boldsymbol{c})}{p_{\mathrm{data}}(\mathbf{x}_t|\boldsymbol{c})} - \gamma\nabla_{\mathbf{x}_t} \log \sigma\left(r\left(\boldsymbol{c}, \mathbf{x}_t\right) - r\left(\boldsymbol{c}, \mathbf{x}_t^l\right)\right)\right\|_2^2 \\
&= \omega(t)\left\|\nabla_{\mathbf{x}_t} \log \frac{p_\theta(\mathbf{x}_t|\boldsymbol{c})}{p_{\mathrm{data}}(\mathbf{x}_t|\boldsymbol{c})} - \gamma(1 - \sigma\left(r\left(\boldsymbol{c}, \mathbf{x}_t\right) - r\left(\boldsymbol{c}, \mathbf{x}_t^l\right)\right))\nabla_{\mathbf{x}_t} r\left(\boldsymbol{c}, \mathbf{x}_t\right)\right\|_2^2.
\end{aligned}
\tag{24}
$$

Then, we use the reward model $r(\mathbf{x}_t, \boldsymbol{c})$ in Equation (11) and get the gradient of the reward:

$$\nabla_{\mathbf{x}_t} r(\mathbf{x}_t, \boldsymbol{c}) = \nabla_{\mathbf{x}_t} \lambda \log \frac{p_\theta(\mathbf{x}_t \mid \mathbf{x}_{t+1}, \boldsymbol{c})}{p_{\text{ref}}(\mathbf{x}_t \mid \mathbf{x}_{t+1}, \boldsymbol{c})}. \tag{25}$$

Then, we get the gradient of $r(\cdot)$ based on the reverse process in Equation (21):

$$\begin{aligned}\nabla_{\mathbf{x}_t} r(\boldsymbol{c}, \mathbf{x}_t) &= \lambda \log \frac{p_\theta(\mathbf{x}_t \mid \mathbf{x}_{t+1}, \boldsymbol{c})}{p_{\text{ref}}(\mathbf{x}_t \mid \mathbf{x}_{t+1}, \boldsymbol{c})} \\ &= -\frac{\lambda}{2\sigma_{t+1}^2} \sqrt{\frac{\alpha_t}{\alpha_{t+1}}} \frac{\beta_{t+1}}{\sqrt{1-\bar{\alpha}_{t+1}}} (\boldsymbol{\epsilon}_\theta(\mathbf{x}_{t+1}, \boldsymbol{c}, t+1) - \boldsymbol{\epsilon}_{\text{ref}}(\mathbf{x}_{t+1}, \boldsymbol{c}, t+1)).\end{aligned} \tag{26}$$

Using the definition of the score function which connects the score model and diffusion models as described in Song et al. (2020), we derive the formula for the first term of Equation 24:

$$\nabla_{\mathbf{x}_t} \log \frac{p_\theta(\mathbf{x}_t \mid \boldsymbol{c})}{p_{\text{data}}(\mathbf{x}_t \mid \boldsymbol{c})} = -\frac{1}{\sqrt{1-\bar{\alpha}_t}}(\boldsymbol{\epsilon}_\theta(\mathbf{x}_{t+1}, \boldsymbol{c}, t+1) - \boldsymbol{\epsilon}_t). \tag{27}$$

By combining Equation (23), (27), we can obtain our final objectives:

$$\begin{aligned}&\min_\theta \omega(t) \left\| \nabla_{\mathbf{x}_t} \log \frac{p_\theta(\mathbf{x}_t \mid \boldsymbol{c})}{p_{\text{data}}(\mathbf{x}_t \mid \boldsymbol{c})} - \gamma \nabla_{\mathbf{x}_t} \log \sigma\left(r(\mathbf{x}_t, \boldsymbol{c}) - r(\mathbf{x}_t^l, \boldsymbol{c})\right) \right\|_2^2 \\ &= \min_\theta A(t) \left\| B(t)(\boldsymbol{\epsilon}_{\theta,t+1} - \boldsymbol{\epsilon}_{t+1}) - \lambda\gamma\left(1 - \sigma\left(r(\boldsymbol{c}, \mathbf{x}_t) - r(\boldsymbol{c}, \mathbf{x}_t^l)\right)\right)(\boldsymbol{\epsilon}_{\theta,t+1} - \boldsymbol{\epsilon}_{\text{ref},t+1}) \right\|_2^2,\end{aligned} \tag{28}$$

where the specific value are $\boldsymbol{\epsilon}_{\theta,t+1} = \boldsymbol{\epsilon}_\theta(\mathbf{x}_{t+1}, \boldsymbol{c}, t+1)$, $A(t) = \omega(t)\frac{1}{4\sigma_{t+1}^4}\frac{\alpha_t}{\alpha_{t+1}}\frac{\beta_{t+1}^2}{1-\bar{\alpha}_{t+1}}$, $B(t) = \omega(t)\frac{1}{2\sigma_{t+1}^2}\sqrt{\frac{\alpha_t}{\alpha_{t+1}}}\frac{\beta_{t+1}\sqrt{1-\bar{\alpha}_t}}{\sqrt{1-\bar{\alpha}_{t+1}}}$ and similarly for $\boldsymbol{\epsilon}_{\text{ref},t+1}$.

## C DERIVATIONS AND PROOF IN SECTION 4.3

### C.1 DERIVATION OF EQUATION (15)

In this section, we provide a detailed derivation of Equation (15). Starting from the RLHF objective on T2I diffusion models in Equation (5), we can have the following equation:

$$\mathbb{E}_{\boldsymbol{c}\sim\mathcal{D}}\mathbb{E}_{p_\theta(\mathbf{x}_{0:T}|\boldsymbol{c})} \sum_{t=0}^{T-1} r(\mathbf{x}_t, \boldsymbol{c}) - \lambda\mathbb{D}_{\text{KL}}\left[p_\theta(\mathbf{x}_{0:T}|\boldsymbol{c}) \| p_{\text{ref}}(\mathbf{x}_{0:T}|\boldsymbol{c})\right] \tag{29}$$

The we can get the following equation:

$$\begin{aligned}&\mathbb{E}_{p_\theta(\mathbf{x}_t|\mathbf{x}_{t+1},\boldsymbol{c})} r(\mathbf{x}_t, \boldsymbol{c}) - \lambda\mathbb{D}_{\text{KL}}\left[p_\theta(\mathbf{x}_t \mid \mathbf{x}_{t+1}, \boldsymbol{c}) \| p_{\text{ref}}(\mathbf{x}_t \mid \mathbf{x}_{t+1}, \boldsymbol{c})\right], \\ &= \mathbb{E}_{p_\theta(\mathbf{x}_t|\mathbf{x}_{t+1},\boldsymbol{c})} \lambda \log e^{\frac{1}{\lambda}r(\mathbf{x}_t,\boldsymbol{c})} - \lambda\mathbb{E}_{p_\theta(\mathbf{x}_t|\mathbf{x}_{t+1},\boldsymbol{c})} \log \frac{p_\theta(\mathbf{x}_t \mid \mathbf{x}_{t+1}, \boldsymbol{c})}{p_{\text{ref}}(\mathbf{x}_t \mid \mathbf{x}_{t+1}, \boldsymbol{c})} \\ &= \mathbb{E}_{p_\theta(\mathbf{x}_t|\mathbf{x}_{t+1},\boldsymbol{c})} \lambda \log \frac{e^{\frac{1}{\lambda}r(\mathbf{x}_t,\boldsymbol{c})}p_{\text{ref}}(\mathbf{x}_t \mid \mathbf{x}_{t+1}, \boldsymbol{c})}{p_\theta(\mathbf{x}_t \mid \mathbf{x}_{t+1}, \boldsymbol{c})}.\end{aligned} \tag{30}$$

We have the analytical form of $p^*(\mathbf{x}_{0:T}|\boldsymbol{c})$ following (Wallace et al., 2024):

$$p^*(\mathbf{x}_{0:T}|\boldsymbol{c}) = p_{\text{ref}}(\mathbf{x}_{0:T}|\boldsymbol{c}) e^{\left(\sum_{t=0}^{T-1} r(\mathbf{x}_t,\boldsymbol{c})\right)/\lambda}/Z(\boldsymbol{c}), \tag{31}$$

where $Z(\boldsymbol{c}) = \int \exp(\sum_{t=0}^{T-1} r(\mathbf{x}_t, \boldsymbol{c})/\lambda)p_{\text{ref}}(\mathbf{x}_{0:T}|\boldsymbol{c})\mathrm{d}\mathbf{x}_{0,T-1}$. Therefore, we can get the following results for Equation (30):

$$\begin{aligned}&\mathbb{E}_{p_\theta(\mathbf{x}_{0:T}|\boldsymbol{c})} \lambda \log \frac{e^{\frac{1}{\lambda}\sum_{t=0}^{T-1} r(\mathbf{x}_t,\boldsymbol{c})}p_{\text{ref}}(\mathbf{x}_{0:T}|\boldsymbol{c})}{p_\theta(\mathbf{x}_{0:T}|\boldsymbol{c})} \\ &= \mathbb{E}_{p_\theta(\mathbf{x}_{0:T}|\boldsymbol{c})} \lambda \log \frac{p^*(\mathbf{x}_{0:T}|\boldsymbol{c}) Z(\boldsymbol{c})}{p_\theta(\mathbf{x}_{0:T}|\boldsymbol{c})} \\ &= -\lambda\mathbb{D}_{\text{KL}}\left[(p_\theta(\mathbf{x}_{0:T}|\boldsymbol{c}) \| p^*(\mathbf{x}_{0:T}|\boldsymbol{c})\right] + \lambda \log Z(\boldsymbol{c}).\end{aligned} \tag{32}$$

We can get the $\mathbb{D}_{\mathrm{KL}}(\cdot)$ form as follows:

$$
\begin{aligned}
\mathbb{D}_{\mathrm{KL}}\left[(p_\theta(\mathbf{x}_{0:T}|\boldsymbol{c})\|p^*(\mathbf{x}_{0:T}|\boldsymbol{c})\right] &= \int p_\theta\left(\mathbf{x}_{0:T}\mid\boldsymbol{c}\right)\times\log\frac{p_\theta\left(\mathbf{x}_{0:T}\mid\boldsymbol{c}\right)}{p^*\left(\mathbf{x}_{0:T}\mid\boldsymbol{c}\right)}d\mathbf{x}_{0:T}\\
&= \int p_\theta\left(\mathbf{x}_{0:T}\mid\boldsymbol{c}\right)\log\frac{p_\theta\left(\mathbf{x}_T\mid\boldsymbol{c}\right)\prod_{t=0}^{T-1}p_\theta\left(\mathbf{x}_t\mid\mathbf{x}_t,\boldsymbol{c}\right)}{p^*\left(\mathbf{x}_T\mid\boldsymbol{c}\right)\prod_{t=0}^{T-1}p^*\left(\mathbf{x}_t\mid\mathbf{x}_{t+1},\boldsymbol{c}\right)}d\mathbf{x}_{0:T}\\
&= \int p_\theta\left(\mathbf{x}_{0:T}\mid\boldsymbol{c}\right)\left(\log\frac{p_\theta\left(\mathbf{x}_T\mid\boldsymbol{c}\right)}{p^*\left(\mathbf{x}_T\mid\boldsymbol{c}\right)}+\sum_{t=0}^{T-1}\log\frac{p_\theta\left(\mathbf{x}_t\mid\mathbf{x}_{t+1},\boldsymbol{c}\right)}{p^*\left(\mathbf{x}_t\mid\mathbf{x}_{t+1},\boldsymbol{c}\right)}\right)d\mathbf{x}_{0:T}\\
&= \sum_{t=1}^{T}\mathbb{E}_{p_\theta(\mathbf{x}_{t+1:T}|\boldsymbol{c})}\mathbb{E}_{p_\theta(\mathbf{x}_{0:t}|\mathbf{x}_{t+1:T},\boldsymbol{c})}\left[\log\frac{p_\theta\left(\mathbf{x}_t\mid\mathbf{x}_{t+1},\boldsymbol{c}\right)}{p^*\left(\mathbf{x}_t\mid\mathbf{x}_{t+1},\boldsymbol{c}\right)}\right]\\
&= \sum_{t=1}^{T}\mathbb{E}_{p_\theta(\mathbf{x}_{t+1}|\boldsymbol{c})}\mathbb{E}_{p_\theta(\mathbf{x}_t|\mathbf{x}_{t+1},\boldsymbol{c})}\left[\log\frac{p_\theta\left(\mathbf{x}_t\mid\mathbf{x}_{t+1},\boldsymbol{c}\right)}{p^*\left(\mathbf{x}_t\mid\mathbf{x}_{t+1},\boldsymbol{c}\right)}\right]\\
&= \sum_{t=1}^{T}\mathbb{E}_{p_\theta(\mathbf{x}_{t+1}|\boldsymbol{c})}\mathbb{D}_{\mathrm{KL}}\left[p_\theta\left(\mathbf{x}_t\mid\mathbf{x}_{t+1},\boldsymbol{c}\right)\|p^*\left(\mathbf{x}_t\mid\mathbf{x}_{t+1},\boldsymbol{c}\right)\right]\\
&= \mathbb{E}_{p_\theta(\mathbf{x}_{0:T}|\boldsymbol{c})}\sum_{t=0}^{T-1}\mathbb{D}_{\mathrm{KL}}\left[p_\theta(\mathbf{x}_t\mid\mathbf{x}_{t+1},\boldsymbol{c})\|p^*(\mathbf{x}_t\mid\mathbf{x}_{t+1},\boldsymbol{c})\right],
\end{aligned}
\tag{33}
$$

where $p^*(\mathbf{x}_t\mid\mathbf{x}_{t+1},\boldsymbol{c})\propto p_{\mathrm{ref}}\left(\boldsymbol{x}_t\mid\boldsymbol{x}_{t+1},\boldsymbol{c}\right)e^{(r(\boldsymbol{x}_t,\boldsymbol{c}))/\lambda}$. Finally, we can combine the above equation with Equation (29):

$$
\mathcal{L}_{\mathrm{rlhf}}=\mathbb{E}_{\boldsymbol{c}\sim\mathcal{D}}\mathbb{E}_{p_\theta(\mathbf{x}_{0:T}|\boldsymbol{c})}\sum_{t=0}^{T-1}-\lambda\mathbb{D}_{\mathrm{KL}}\left[p_\theta(\mathbf{x}_t\mid\mathbf{x}_{t+1},\boldsymbol{c})\|p^*(\mathbf{x}_t\mid\mathbf{x}_{t+1},\boldsymbol{c})\right]+\lambda\log Z\left(\boldsymbol{c}\right).
\tag{34}
$$

### C.2 PROOF OF THEOREM 1

**Theorem 1.** *Following $\omega(t)=2\sigma_{t+1}^2/\lambda$, $\gamma=1/2\lambda$, reward model $r(\cdot)$ as defined in Equation (4) and $p_{\mathrm{data}}(\cdot)$ as the reference model for RLHF of T2I diffusion models in Equation (15), the gradient of DSPO objective in Equation (13) by sampling data from $p_\theta$ satisfies:*

$$
\nabla_\theta\mathcal{L}_{\mathrm{rlhf}}=\nabla_\theta\mathbb{E}_{\boldsymbol{c}\sim\mathcal{D}}\mathbb{E}_{p_\theta(\mathbf{x}_{0:T}|\boldsymbol{c})}\sum_{t=0}^{T-1}-\mathcal{L}_{\mathrm{DSPO}}^t.
\tag{35}
$$

First, we recognize the reverse process of diffusion models as a Gaussian process, which we denote as $p_\theta\left(\boldsymbol{x}_t\mid\boldsymbol{x}_{t+1},\boldsymbol{c}\right)\sim\mathcal{N}(\boldsymbol{\mu},\boldsymbol{\Sigma})$ for simplicity. The corresponding $\log p_\theta(\cdot)$ can then be defined as:

$$
\log p_\theta\left(\mathbf{x}_t\mid\mathbf{x}_{t+1}\right)=-\frac{1}{2}\left(\mathbf{x}_t-\boldsymbol{\mu}\right)^T\boldsymbol{\Sigma}^{-1}\left(\mathbf{x}_t-\boldsymbol{\mu}\right)+C,
\tag{36}
$$

where $C$ is a constant value and we can denote $p_{\mathrm{ref}}\left(\boldsymbol{x}_t\mid\boldsymbol{x}_{t+1},\boldsymbol{c}\right)$ by replacing $\mu$ with $\mu_{\mathrm{ref}}$ similarly. $\log p^*\left(\mathbf{x}_t\mid\mathbf{x}_{t+1},\boldsymbol{c}\right)$ can be estimated by a Taylor expansion around $\mathbf{x}_t$:

$$
\begin{aligned}
\log p^*\left(\mathbf{x}_t\mid\mathbf{x}_{t+1},\boldsymbol{c}\right) &= \log p_{\mathrm{ref}}\left(\boldsymbol{x}_t\mid\boldsymbol{x}_{t+1},\boldsymbol{c}\right)e^{(r(\boldsymbol{x}_t,\boldsymbol{c}))/\lambda}/Z(\mathbf{x}_{t+1},\boldsymbol{c})\\
&\approx \log p_{\mathrm{ref}}\left(\boldsymbol{x}_t\mid\boldsymbol{x}_{t+1},\boldsymbol{c}\right)+\log e^{(r(\boldsymbol{x}_t,\boldsymbol{c}))/\lambda}\Big|_{\mathbf{x}_t=\boldsymbol{\mu}}+\left(\mathbf{x}_t-\boldsymbol{\mu}_{\mathrm{ref}}\right)\nabla_{\mathbf{x}_t}\log e^{(r(\mathbf{x}_t,\boldsymbol{c}))/\lambda}\Big|_{\mathbf{x}_t=\boldsymbol{\mu}_{\mathrm{ref}}}-C_1\\
&= -\frac{1}{2}\left(\mathbf{x}_t-\boldsymbol{\mu}_{\mathrm{ref}}\right)^T\boldsymbol{\Sigma}^{-1}\left(\mathbf{x}_t-\boldsymbol{\mu}_{\mathrm{ref}}\right)+\left(\mathbf{x}_t-\boldsymbol{\mu}_{\mathrm{ref}}\right)\nabla_{\mathbf{x}_t}\frac{r\left(\mathbf{x}_t,\boldsymbol{c}\right)}{\lambda}+C_2\\
&= -\frac{1}{2}\left(\mathbf{x}_t-\boldsymbol{\mu}_{\mathrm{ref}}-\boldsymbol{\Sigma}\nabla_{\mathbf{x}_t}\frac{r\left(\mathbf{x}_t,\boldsymbol{c}\right)}{\lambda}\right)^T\boldsymbol{\Sigma}^{-1}\left(\mathbf{x}_t-\boldsymbol{\mu}_{\mathrm{ref}}-\boldsymbol{\Sigma}\nabla_{\mathbf{x}_t}\frac{r\left(\mathbf{x}_t,\boldsymbol{c}\right)}{\lambda}\right)+C_3\\
&= \log p^*(\mathbf{a})+C_4, \mathbf{a}\sim\mathcal{N}(\boldsymbol{\mu}_{\mathrm{ref}}+\boldsymbol{\Sigma}\nabla_{\mathbf{x}_t}\frac{r\left(\mathbf{x}_t,\boldsymbol{c}\right)}{\lambda},\boldsymbol{\Sigma}),
\end{aligned}
\tag{37}
$$

where $C_3=-\mathbf{g}^T\boldsymbol{\Sigma}\mathbf{g}/2$ and $\mathbf{g}$ represents $\nabla_{\mathbf{x}_t}r\left(\mathbf{x}_t,\boldsymbol{c}\right)/\lambda$. We can safely ignore the constant term $C_4$, since it corresponds to the normalizing coefficient following Dhariwal & Nichol (2021). Then, we can observe that $p^*\left(\mathbf{x}_t\mid\mathbf{x}_{t+1},\boldsymbol{c}\right)$ follows the Guassian distribution. Therefore, we can further derive the KL divergence loss betwenn two Guassian distributions for RLHF in Equation (15):

$$
\begin{aligned}
&\mathbb{D}_{\mathrm{KL}}\left(p_\theta(\mathbf{x}_t\mid\mathbf{x}_{t+1},\boldsymbol{c})\|p^*(\mathbf{x}_t\mid\mathbf{x}_{t+1},\boldsymbol{c})\right)\\
&= \left(\boldsymbol{\mu}-\boldsymbol{\mu}_{\mathrm{ref}}-\boldsymbol{\Sigma}\nabla_{\mathbf{x}_t}\frac{r\left(\mathbf{x}_t,\boldsymbol{c}\right)}{\lambda}\right)\boldsymbol{\Sigma}^{-1}\left(\boldsymbol{\mu}-\boldsymbol{\mu}_{\mathrm{ref}}-\boldsymbol{\Sigma}\nabla_{\mathbf{x}_t}\frac{r\left(\mathbf{x}_t,\boldsymbol{c}\right)}{\lambda}\right)^T.
\end{aligned}
\tag{38}
$$

We then put the expression of $\boldsymbol{\mu}$, $\boldsymbol{\mu}_{\text{ref}}$ and $\boldsymbol{\Sigma}$ into the above equation:

$$
\begin{aligned}
&\mathbb{D}_{\text{KL}}\left(p_\theta(\mathbf{x}_t \mid \mathbf{x}_{t+1}, \boldsymbol{c}) \| p^*(\mathbf{x}_t \mid \mathbf{x}_{t+1}, \boldsymbol{c})\right) \\
&= \frac{1}{2\sigma_{t+1}^2} \left\| \sqrt{\frac{\alpha_t}{\alpha_{t+1}}} \frac{\beta_{t+1}}{\sqrt{1-\bar{\alpha}_{t+1}}} \left(\boldsymbol{\epsilon}_{\text{ref},t+1} - \boldsymbol{\epsilon}_{\theta,t+1}\right) - \frac{\sigma_{t+1}^2}{\lambda} \nabla_{\mathbf{x}_t} r(\mathbf{x}_t) \right\|_2^2.
\end{aligned}
\tag{39}
$$

We put Equation (27) into our loss function to get:

$$
\begin{aligned}
\mathcal{L}_{\text{DSPO}}^t &= \omega(t) \left\| \frac{1}{2\sigma_{t+1}^2} \sqrt{\frac{\alpha_t}{\alpha_{t+1}}} \frac{\beta_{t+1}}{\sqrt{1-\bar{\alpha}_{t+1}}} \left(\boldsymbol{\epsilon}_{t+1} - \boldsymbol{\epsilon}_\theta\right) - \gamma \nabla_{\mathbf{x}_t} \log \sigma\left(r\left(\mathbf{x}_t, \boldsymbol{c}\right) - r\left(\mathbf{x}_t^l, \boldsymbol{c}\right)\right) \right\|_2^2 \\
&= \frac{\omega(t)}{4\sigma_{t+1}^4} \left\| \sqrt{\frac{\alpha_t}{\alpha_{t+1}}} \frac{\beta_{t+1}}{\sqrt{1-\bar{\alpha}_{t+1}}} \left(\boldsymbol{\epsilon}_{t+1} - \boldsymbol{\epsilon}_\theta\right) - 2\sigma_{t+1}^2 \gamma \nabla_{\mathbf{x}_t} \log \sigma\left(r\left(\mathbf{x}_t, \boldsymbol{c}\right) - r\left(\mathbf{x}_t^l, \boldsymbol{c}\right)\right) \right\|_2^2.
\end{aligned}
\tag{40}
$$

Therefore, we can observe that $\mathcal{L}_{\text{DSPO}}^t = \mathbb{D}_{\text{KL}}\left(p_\theta(\mathbf{x}_t \mid \mathbf{x}_{t+1}, \boldsymbol{c}) \| p^*(\mathbf{x}_t \mid \mathbf{x}_{t+1}, \boldsymbol{c})\right)$ when $\omega(t) = 2\sigma_{t+1}^2/\lambda$, $\gamma = 1/2\lambda$ and we use $p_{\text{data}}(\cdot)$ as the reference model, we get the following equation:

$$
\mathcal{L}_{\text{rlhf}} = \mathbb{E}_{\boldsymbol{c} \sim \mathcal{D}} \mathbb{E}_{p_\theta(\mathbf{x}_{0:T}|\boldsymbol{c})} \sum_{t=0}^{T-1} -\mathcal{L}_{\text{DSPO}}^t + \lambda \log Z\left(\boldsymbol{c}\right).
\tag{41}
$$

Finally, it completes our proof:

$$
\nabla_\theta \mathcal{L}_{\text{rlhf}} = \nabla_\theta \mathbb{E}_{\boldsymbol{c} \sim \mathcal{D}} \mathbb{E}_{p_\theta(\mathbf{x}_{0:T}|\boldsymbol{c})} \sum_{t=0}^{T-1} -\mathcal{L}_{\text{DSPO}}^t
\tag{42}
$$

## C.3 DERIVATION OF LOSS WITH REWARD MODEL OF RLHF

Through the proof in Section C.2, we can easily get the conclusion that if we set $p\left(\mathbf{y} \mid \mathbf{x}_t, \boldsymbol{c}\right) = p\left(\mathbf{y} \mid \mathbf{x}_t, \boldsymbol{c}\right) = \exp\left(r(\mathbf{x}_t, \boldsymbol{c})/\lambda\right)/Z(\boldsymbol{c})$, the direction of minimizing our proposed loss is same as maximizing $\mathcal{L}_{\text{rlhf}}$ when the hyperparameters of both losses are properly adjusted. Specifically, we can get the optimization objective with the above format of $p\left(\mathbf{y} \mid \mathbf{x}_t, \boldsymbol{c}\right)$:

$$
\min_\theta \omega(t) \left\| \nabla_{\mathbf{x}_t} \log \frac{p_\theta(\mathbf{x}_t \mid \boldsymbol{c})}{p_{\text{data}}(\mathbf{x}_t \mid \boldsymbol{c})} - \gamma \nabla_{\mathbf{x}_t} r\left(\mathbf{x}_t, \boldsymbol{c}\right) \right\|_2^2
\tag{43}
$$

Referring to the proof in Appendix B.2 and disregarding some weighting parameters, we derive the following equation for the energy-based classifier:

$$
\begin{aligned}
\mathcal{L}_{\text{DSPO-E}}^t &= \omega(t) \left\| \frac{1}{2\sigma_{t+1}^2} \sqrt{\frac{\alpha_t}{\alpha_{t+1}}} \frac{\beta_{t+1}}{\sqrt{1-\bar{\alpha}_{t+1}}} \left(\boldsymbol{\epsilon}_{t+1} - \boldsymbol{\epsilon}_\theta\right) - \gamma \nabla_{\mathbf{x}_t} r\left(\mathbf{x}_t, \boldsymbol{c}\right) \right\|_2^2 \\
&= \omega(t) \frac{1}{4\sigma_{t+1}^4} \left\| \sqrt{\frac{\alpha_t}{\alpha_{t+1}}} \frac{\beta_{t+1}}{\sqrt{1-\bar{\alpha}_{t+1}}} \left(\boldsymbol{\epsilon}_{t+1} - \boldsymbol{\epsilon}_\theta\right) - 2\sigma_{t+1}^2 \gamma \nabla_{\mathbf{x}_t} r\left(\mathbf{x}_t, \boldsymbol{c}\right) \right\|_2^2
\end{aligned}
\tag{44}
$$

Therefore, we can obtain the equivalent form of Equation (39) with $\omega(t) = 2\sigma(t)^2/\lambda$, $\gamma = 1/2\lambda$ and we use $p_{\text{data}}(\cdot)$ as the reference model. Based on this formulation, we can get the following objective when considering $r(\cdot)$ as defined in Equation (11):

$$
\begin{aligned}
\mathcal{L}_{\text{DSPO-E}}^t &= \omega(t) \left\| \nabla_{\mathbf{x}_t} \log \frac{p_\theta(\mathbf{x}_t|\mathbf{x}_{t+1}, \boldsymbol{c})}{p_{\text{data}}(\mathbf{x}_t|\mathbf{x}_{t+1}, \boldsymbol{c})} - \gamma \nabla_{\mathbf{x}_t} r\left(\mathbf{x}_t, \boldsymbol{c}\right) \right\|_2^2 \\
&= A(t) \left\| \boldsymbol{\epsilon}_{\theta,t+1} - \boldsymbol{\epsilon}_{t+1} - \gamma\beta \left(\boldsymbol{\epsilon}_{\theta,t+1} - \boldsymbol{\epsilon}_{\text{ref},t+1}\right) \right\|_2^2,
\end{aligned}
\tag{45}
$$

where $A(t) = \omega(t) \frac{1}{2\sigma_{t+1}^2} \frac{\alpha_t}{\alpha_{t+1}} \frac{\beta_{t+1}^2}{1-\bar{\alpha}_{t+1}}$, $\boldsymbol{\epsilon}_{\theta,t+1} = \boldsymbol{\epsilon}_\theta\left(\mathbf{x}_{t+1}, \boldsymbol{c}, t+1\right)$ and similarly for $\boldsymbol{\epsilon}_{\text{ref},t+1}$. The detailed derivation are the same to the derivation of Equation (12), which are shown in Section B.2.

# D EXPERIMENTAL DETAILS

## D.1 THE DETAILS OF TRAINING ON SDXL MODELS

Inspired by the observation from Diffusion-DPO (Wallace et al., 2024), which highlights that when the quality of training data is lower than that of data generated by the original model, using the

reference model becomes a preferable choice. Therefore, we incorporate the reference model in Equation (10) and derive the following equation for training the SDXL models from Equation (12):

$$\min_{\theta} \frac{A(t)}{\sqrt{\lambda\gamma}} \left\| \left( 1 - \frac{B(t)}{\lambda\gamma} - \sigma \left( r\left(\boldsymbol{c}, \mathbf{x}_t\right) - r\left(\boldsymbol{c}, \mathbf{x}_t^l\right) \right) \right) \left( \boldsymbol{\epsilon}_{\theta,t+1} - \boldsymbol{\epsilon}_{\text{ref},t+1} \right) \right\|_2^2, \qquad (46)$$

where we replace the first term $\boldsymbol{\epsilon}_{t+1}$ with $\boldsymbol{\epsilon}_{\text{ref},t+1}$. In practical implementation, we observe that the term $(\boldsymbol{\epsilon}_{\theta,t+1} - \boldsymbol{\epsilon}_{\text{ref},t+1})$ is small, yet it significantly impacts the training speed. Additionally, using small values of $B(t)/(\lambda\gamma)$ yields good performance. Therefore, to simplify the training process of SDXL and reduce hyperparameter, we train the following loss for SDXL:

$$\min_{\theta} \left\| 1 - \sigma \left( r\left(\boldsymbol{c}, \mathbf{x}_t\right) - r\left(\boldsymbol{c}, \mathbf{x}_t^l\right) \right) \right\|_2^2. \qquad (47)$$

Then, we replace $r\left(\boldsymbol{c}, \mathbf{x}_t\right)$ and $r\left(\boldsymbol{c}, \mathbf{x}_t^l\right)$ as discussed in Appendix B and ignore the relevant part for $\alpha_t$ and $\beta_t$ as the training process of diffusion models.

## D.2 THE DETAILS OF DATASETS

In this section, we provide detailed descriptions of datasets:

**Pick-a-Pic v2** (Pick V2) (Kirstain et al., 2023): The Pick-a-Pic dataset was developed by logging user interactions with the Pick-a-Pic web application for text-to-image generation. It contains over 500,000 examples and 35,000 distinct prompts. Each example includes a prompt, two generated images, and a label indicating which image is preferred or if there is no significant preference (tie). The dataset was generated using multiple backbone models, including Stable Diffusion 2.1, Dreamlike Photoreal 2.05, and Stable Diffusion XL variants (Rombach et al., 2022), with varying classifier-free guidance scale values (Ho & Salimans, 2022).

**Parti-Prompts** (Yu et al., 2022): Parti-Prompts is a comprehensive dataset consisting of over 1,600 prompts written in English, designed to evaluate and benchmark the capabilities of text-to-image generation models. These prompts span a wide range of categories, offering a diverse set of challenges to assess model performance across various dimensions.

**HPSV2** (Wu et al., 2023): The dataset includes a total of 98,807 images generated from 25,205 unique prompts. For each prompt, multiple images are generated, with one image selected by the user as the preferred choice while the others serve as non-preferred negatives. The number of images per prompt varies, with 23,722 prompts having four images, 953 prompts having three images, and 530 prompts having two images.

**InstructPix2Pix** (Brooks et al., 2023): InstructPix2Pix is a dataset designed to edit images based on human-provided instructions. For example, with a prompt like "make the clouds rainy," the model will modify the input image accordingly. It conditions its output on both the text prompt (editing instruction) and the input image, enabling intuitive, instruction-driven image edits. We conducted our image editing experiment on 1,000 test samples from this dataset [1].

## D.3 IMPLEMENTATION DETAILS

We present implementation and setup details of DSPO in this section. For experiments, we use the AdamW optimizer with an effective batch size of 2048 pairs, as outlined in Wallace et al. (2024). Training is conducted on 4 NVIDIA V100 GPUs, with a local batch size of 4 pair and gradient accumulation over 128 steps. We train at fixed square resolutions and use a learning rate $2.048 \cdot 10^{-8}$, scheduled with a 2000-step linear warmup, followed by inverse scaling (Rafailov et al., 2024). We present the DSPO results with $\lambda = 0.001$. For a fair comparison, we use the default hyperparameters for the T2I diffusion model with the image editing task with text instructions, ensuring consistency in evaluation, i.e., guidance scale as 7.5 and the strength as 0.75. Our code of DSPO is based on the implementation Diffusion-DPO [2].

---

[1] https://huggingface.co/datasets/fusing/instructpix2pix-1000-samples
[2] https://github.com/SalesforceAIResearch/DiffusionDPO

Table 6: Win-rate (VS SD15) comparison of PickV2 dataset for Ablation Studies.

| Dataset | Method | Pick Score | HPS | Aesthetics | CLIP | Image Reward |
|---|---|---|---|---|---|---|
| PickV2 | DSPO-ref | 65.40 | 75.00 | 68.00 | 57.20 | 64.40 |
| | DSPO-nodup | 71.20 | 82.60 | 74.00 | 58.60 | 76.60 |
| | DSPO-LoRA | 68.00 | 79.40 | 74.40 | 60.20 | 70.80 |
| | DSPO | **73.60** | **84.80** | **76.20** | **61.80** | **78.00** |

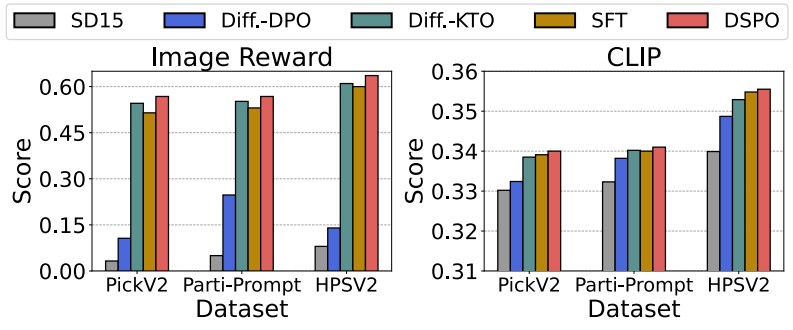

Figure 7: Reward score results for with Image Reward and CLIP models on all dataset.

# E    ADDITIONAL EXPERIMENT RESULTS

## E.1    ADDITIONAL ABLATION STUDIES AND OTHER EXPERIMENTS

In this section, we conduct the ablation studies by using $p_{\text{ref}}$ in Equation (10), denoted as DSPO-ref. Additionally, since the PickV2 dataset contains many duplicate prompts, we conduct further experiments by removing these duplicates and rerunning our model, denoted as DSPO-nodup. To reduce memory usage and computational time, we fine-tuned SD15 using LoRA combined with DSPO, referred to as DSPO-LoRA. The results of these three experiments are summarized in a single table, as presented in Table 6. We get the following observation: (i) our method, which leverages the score function of the true data distribution, outperforms the approach (DSPO-ref) that treats the original pretrained model as the reference model. This is because the quality of images generated by the original pretrained stable diffusion models (SD 1.5) is lower than that of the fine-tuning dataset. As a result, using the score function of the true data distribution can produce better outcomes; (ii) DSPO performs better compared with DSPO-nodup. It means that randomly dropping duplicate samples without careful selection may have negatively impacted the results, potentially leading to the loss of important information by randomly retaining only one instance of a prompt and discarding the others; (iii) fine-tuning with LoRA enhances performance, though it slightly falls short of achieving results similar to full parameter training. However, the performance gap between these two methods is minimal, making LoRA a viable approach for fine-tuning large models.

## E.2    ADDITIONAL REWARD SCORE RESULTS

In this section, we present additional reward score results using CLIP and Image Reward, as illustrated in Figure 7. Consistent with previous findings, our model surpasses all baseline methods, further demonstrating the effectiveness of DSPO.

## E.3    ADDITIONAL RESULTS FOR QUALITATIVE ANALYSIS

To further verify the effectiveness of our model, we provide more Qualitative results of Text to image generation for different baselines in Figure 8. We list the prompts used in Figure 8 as follows:

1. Minotaur

2. a painting of a fox in the style of starry night

3. The image is a vibrant and intricate illustration of a man, with a focus on his shoulder and head, created using inkpen and Unreal Engine technology.

4. a portrait of young girl

5. A head shot of a pretty girl dressed in a cyberpunk version of Marie Antoinette's rococo style, depicted through detailed digital art and trending on Art Station.

6. Nine human faces from Neanderthal to Modern Human and beyond depict the future of human appearance.

7. A digital painting of a young pirate with sharp features and a piercing gaze.

8. Cyberpunk cat.

Moreover, we also provide more qualitative results of the image editing task with text instructions in Figure 9. Similarly, we list the prompts used in Figure 9:

1. make it marble

2. A fantasy landscape, trending on artstation

3. turn it into a painting

4. make it a seascape

5. make the cub a tiger

6. turn it into a computer game

7. As an oil painting

## F   ETHICAL STATEMENT

This study explores new algorithms to fine-tune text-to-image diffusion models for human preference alignment. We use public data following the prior works (Wallace et al., 2024) in the field of human preference alignment on text-to-image generation for both training and evaluation, which can be directly downloaded from Hugging Face. Moreover, no sensitive user information is exposed, and all experimental results are presented as aggregate statistics to maintain reproducibility without risking information leakage. These practices comply with ethical and legal standards, ensuring a responsible approach to AI research.

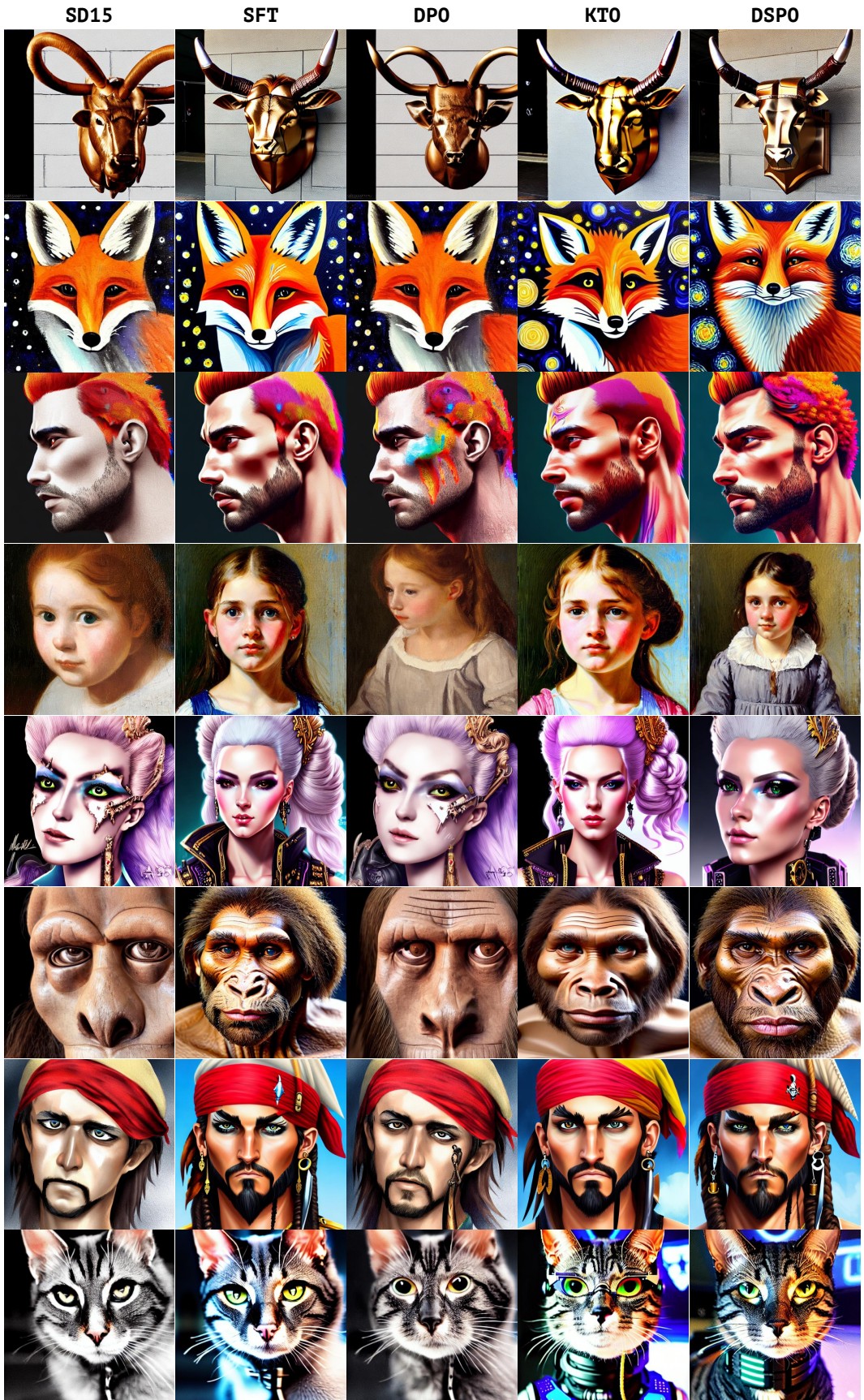

Figure 8: Images generated by different models for various prompts which are selected from PickV2, Parti-Prompt and HPSV2. Detailed prompts for these images are provided in Section E.3.

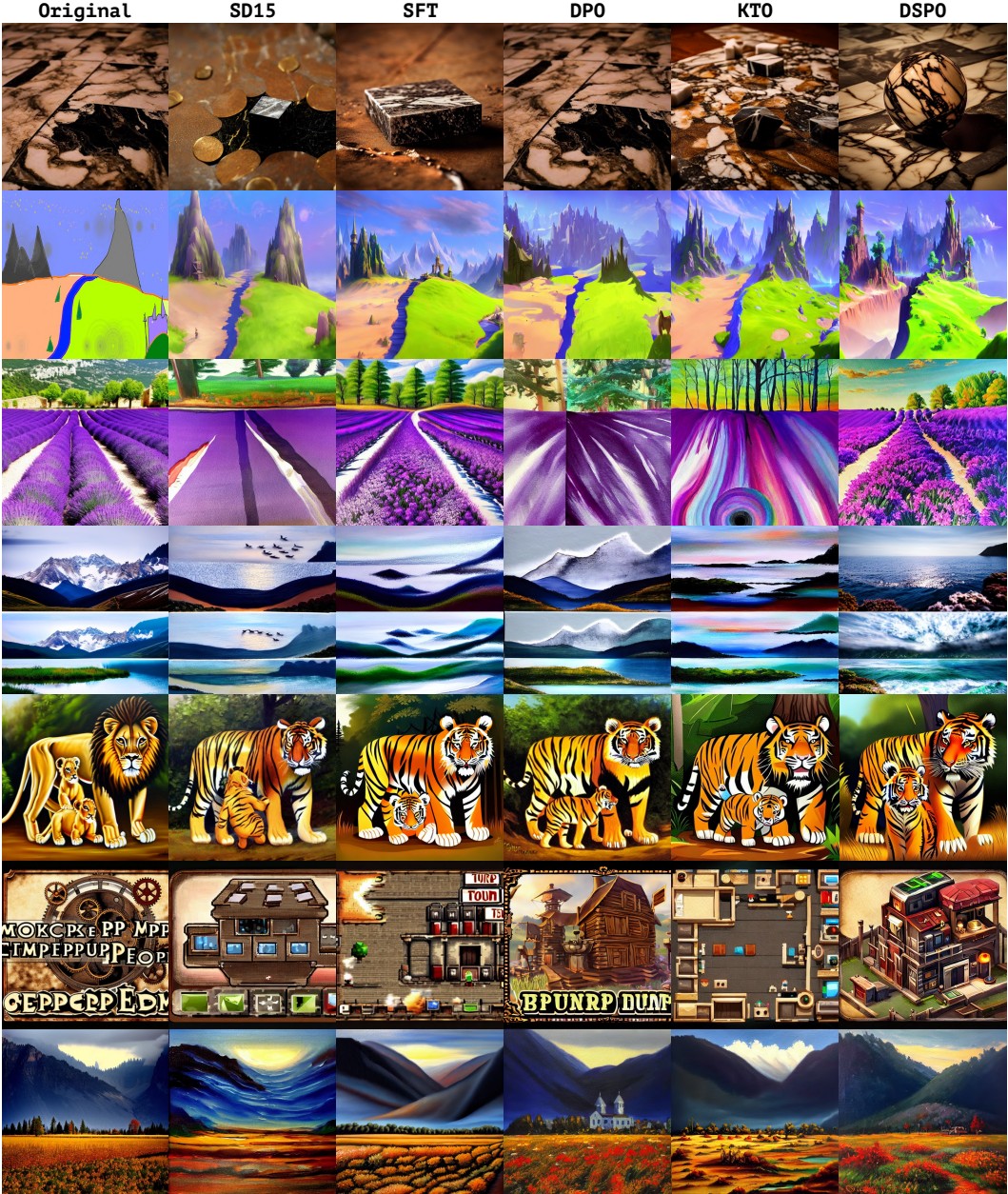

Figure 9: Images generated by different models for various prompts which are selected from InstructPix2Pix of text-guided editing. Detailed prompts for these images are provided in Section E.3.

