# OpenReview forum: "DSPO: Direct Score Preference Optimization for Diffusion Model Alignment"
_ICLR.cc/2025/Conference — ICLR 2025 Oral_

### Official Review · Reviewer_gpAu · 2024-10-30

**Soundness:** 3
**Presentation:** 2
**Contribution:** 3
**Rating:** 6
**Confidence:** 5

**Summary:**

The paper introduces DSPO, presenting a score-matching formulation for fine-tuning pre-trained diffusion models on human preference data. The authors argue that since existing preference alignment fine-tuning methods have a different objective than the pre-training objective, it can lead to sub-optimal results, and they demonstrate this with empirical evidence.

**Strengths:**

* The score-matching formulation for alignment fine-tuning of diffusion models hasn't been explored before, and the paper does a good job of exploring this direction.
* The connection between the objective covered by the RLHF methods for diffusion models and DSPO.

**Weaknesses:**

* The paper misses out on using MaPO [1] as a reasonable baseline even though it considers contemporary works like Diffusion KTO. The reason why I think MaPO is important here to consider is because it has similar motivations and also either performs on par with Diffusion DPO or outperforms it under various settings.
* Lack of experimental results on models like SDXL makes it unclear as to how scalable DSPO is and if it works for models other than SD v1.5.
* The ablations lack experiments on some of the design choices the authors make to arrive at the final objective of DSPO. For example, they use the direct score function of the underlying data distribution as opposed to using that of $p_{ref}$, but they don't justify it with sufficient experimental results.
* Pick-a-Pic v2 contains duplicate prompts. Did the authors perform any de-duplication? If not, I think it might be better to run at least a few experiments with de-duplication to check if this improves the results.

**References**

[1] Margin-aware Preference Optimization for Aligning Diffusion Models without Reference; Hong et al.; 2024.

**Questions:**

* Figure 2 could mention the base model on which the respective methods were applied.

* L099 - L101: The authors mention "... with existing baselines for preference learning in T2I diffusion models." However, Figure 2 compares the performance of a single base model on which the respective methods were applied. So, I think it's better to be specific and mention the base model in the statement.

* Equation 12 could benefit from an expansion of the notations used. For example, I don't know where $\lambda$ is coming from. Furthermore, it'd be beneficial to highlight the score function of the data distribution replacing $p_{ref}$.

* It's not clear how DSPO incorporates $\mathbf{x}_t^w$. Under Section 4.2, $\mathbf{x}_t^w$ only appears in Equation 14.

* L091: Typo on "constraints".

* SD1.5 is a relatively old model. Since DSPO doesn't consider other recent models like SDXL, SD3, Flux, etc., it's unclear as to how well DSPO generalizes. I can understand that providing further results on SD3 or Flux might be computationally challenging, but I request that the authors at least consider SDXL experiments. Additionally, LoRA fine-tuning (similar to how DPOK [1] does it) when doing DSPO for larger models like SD3 and Flux might help them quickly evaluate its potential better.

* Are there any sample-efficient aspects of DSPO? More specifically, I am interested to see if using the score-matching perspective of alignment fine-tuning like DSPO does can improve alignment with fewer samples than other methods.

* The authors could also consider using human-benchmark arenas such as imgsys [2] for evaluation.

* To assess the practical aspects of DSPO, it would be useful to report the wall-clock time and memory requirements of DSP and compare them against the existing methods.

**References**

[1] DPOK: Reinforcement Learning for Fine-tuning Text-to-Image Diffusion Models; Fan et al.; 2023.

[2] imgsys; fal.ai team.

**Details Of Ethics Concerns:**

It might be better to filter out the Pick-a-Pic dataset to discard the NSFW images.

---

> ### Author Response · Authors · 2024-11-18
> **Response to Reviewer gpAu (part 1/3)**
>
> We sincerely thank the reviewer for the comments and suggestions. Please see our clarifications below:
>
> **Q1. The paper misses out on using MaPO [1] as a reasonable baseline even though it considers contemporary works like Diffusion KTO.**
>
> **A1.** Thanks for your suggestions! This is indeed a relevant paper, and we have cited it in our revised manuscript. We have included the baseline provided in the MaPO trained on the Pick-2-pic dataset (we report win rate by treating SDXL as the base model). Also, We fine-tune SDXL using our proposed DSPO method and get the following win rate (VS SDXL) results:
>
> |     |       |       |    | PickV2         |       |  |
> |-----------|----------|-------|-------|-------|-------|-------|
> |           | Pick Score  | HPS  | Aesthetics | CLIP | Image Reward | Average Results |
> | MaPO   |  54.40  | 69.60  | **68.20** | 51.20 |  61.40 | 59.86 |
> | DSPO   |   **74.00**  | **80.00**  | 54.20 | **59.60** | **68.60**  | **67.28** |
>
> |     |       |       |   |    Parti-Prompt   |       |  |
> |-----------|----------|-------|-------|-------|-------|-------|
> |           | Pick Score  | HPS  | Aesthetics | CLIP | Image Reward | Average Results |
> | MaPO   |  58.34  | 66.54  | **68.23** | 47.43 |  58.64 | 59.84 |
> | DSPO   |  **67.46**  | **81.80**  | 57.84  | **55.02** | **73.47** | **67.12** |
>
>
> |     |       |       ||    HPSV2     |       |  |
> |-----------|----------|-------|-------|-------|-------|-------|
> |           | Pick Score  | HPS  | Aesthetics | CLIP | Image Reward | Average Results |
> | MaPO   |  59.62  | 77.90  | **62.31** | 50.90 |  62.09 | 62.56 |
> | DSPO   | **72.59**   | **83.47**  | 51.41 |  **57.34** | **70.09** |  **66.98** |
>
> Note that we utilize a default guidance scale of 7.5 and 50 sampling steps for these two models with seed 0. The results above indicate that our models outperform MaPO across most reward models, with the exception of aesthetics. **In the last column, we present the average results across different reward models, demonstrating that our model outperforms MaPO. This also demonstrates the effectiveness of our model.**
>
> **Q2. Lack of experimental results on models like SDXL makes it unclear as to how scalable DSPO is and if it works for models other than SD v1.5.**
>
> **A2.** Thanks for your comments! We are keen to extend our methods to additional models. We have finished expanding our approach to SDXL, as you mentioned. Here are the corresponding results for win rate (SDXL as base model):
>
> |     |       |       |PickV2   |       |       |
> |-----------|----------|-------|-------|-------|-------|
> |           | Pick Score  | HPS  | Aesthetics | CLIP | Image Reward |
> | SFT   |  20.80  | 40.60  | 23.20 | 44.80 |  34.40 |
> | DPO   | **75.20**    | 76.20  | 54.10 | 59.40 | 65.20 |
> | DSPO   |  74.00  | **80.00**  | **54.20** | **59.60** | **68.60**  |
>
> |     |       |       |Parti-Prompt  |       |       |
> |-----------|----------|-------|-------|-------|-------|
> |           | Pick Score  | HPS  | Aesthetics | CLIP | Image Reward |
> | SFT   |  17.03 | 33.02 | 27.81 | 36.58 | 37.18  |
> | DPO   | 65.44    | 74.08  | 56.86 | **60.54** | 66.85 |
> | DSPO   |  **67.46**  | **81.80**  | **57.84** | 55.02 | **73.47** |
>
> |     |       |       |HPSV2  |       |       |
> |-----------|----------|-------|-------|-------|-------|
> |           | Pick Score  | HPS  | Aesthetics | CLIP | Image Reward |
> | SFT   | 18.18  | 45.28  | 26.72 | 39.13 |  47.22  |
> | DPO   |  70.31  | 80.81 | 50.78 | **59.31** | 68.75 |
> | DSPO   | **72.59**   | **83.47**  | **51.41** |  57.34 | **70.09** |
>
> These results show that our models can still outperform baselines and verify the effectiveness of our model.
>
> **Q3. The ablations lack experiments on some of the design choices the authors make to arrive at the final objective of DSPO.**
>
> **A3.** Thanks for your suggestions! We conduct this ablation study on treating the pretrained diffusion model $p_{ref}$ and $p_{data}$ as the reference model to fine-tune SD15, which are denoted as DSPO-ref, DSPO. Here are the corresponding results for win rate (SD15 as the base model):
>
> |     |       |       |PickV2   |       |       |
> |-----------|----------|-------|-------|-------|-------|
> |           | Pick Score  | HPS  | Aesthetics | CLIP | Image Reward |
> | DPO-ref   | 65.40  | 75.00 | 68.00 | 57.20  | 64.40 |
> | DSPO   |  **73.60** | **84.80**  | **76.20** | **61.80** | **78.00**  |
>
>
> We observe that our method, which leverages the score function of the true data distribution, **outperforms the approach that treats the original pretrained model as the reference model.** This is because the quality of images generated by the original pretrained stable diffusion models is lower than that of the fine-tuning dataset. As a result, using the score function of the true data distribution can produce better outcomes.

---

> > ### Author Response · Authors · 2024-11-18
> > **Response to Reviewer gpAu (part 2/3)**
> >
> > **Q4. Pick-a-Pic v2 contains duplicate prompts. Did the authors perform any de-duplication?**
> >
> > **A4.** Thanks for your questions! For consistency and fair comparison, we followed the Diffusion-DPO settings and did not remove duplicate prompts in the results. We removed duplicate prompts in Pick-a-Pic v2 and ran DSPO to fine-tune SD15 (denoted as DSPO-nodup), yielding the following results for win rate (SD15 as the base model):
> >
> >
> > |     |       |       |PickV2   |       |       |
> > |-----------|----------|-------|-------|-------|-------|
> > |           | Pick Score  | HPS  | Aesthetics | CLIP | Image Reward |
> > | DSPO-nodup   | 71.20  | 82.60  |  74.00 |   58.60   | 76.60 |
> > | DSPO   |   **73.60**  | **84.80**  | **76.20** | **61.80** | **78.00**  |
> >
> > |     |       |       |Parti-Prompt  |       |       |
> > |-----------|----------|-------|-------|-------|-------|
> > |           | Pick Score  | HPS  | Aesthetics | CLIP | Image Reward |
> > | DSPO-nodup   |  62.43  |  85.33 | 75.12 | 51.85 |  68.81 |
> > | DSPO   |  **65.32**  | **87.50**  | **76.96**  | **54.86** | **71.75** |
> >
> > |     |       |       |HPSV2  |       |       |
> > |-----------|----------|-------|-------|-------|-------|
> > |           | Pick Score  | HPS  | Aesthetics | CLIP | Image Reward |
> > | DSPO-nodup   |  76.53   | 90.53  | 79.18 |  | 78.40 |
> > | DSPO   | **79.90**   | **92.56**  | **80.59** |  **61.13** | **82.31** |
> >
> > We observed that randomly dropping duplicate samples without careful selection may have negatively impacted the results, potentially leading to the loss of important information by randomly retaining only one instance of a prompt and discarding the others. Additionally, we used the hyperparameters optimized for fine-tuning SD15 on the full dataset, but training on a deduplicated dataset may require different hyperparameter settings. We will further explore this in the future work.
> >
> > **Q5. Figure 2 could mention the base model on which the respective methods were applied. I think it's better to be specific and mention the base model in the statement in Figure 2.**
> >
> > **A5.** Thanks for your suggestions! We have added it in the revised paper.
> >
> > **Q6. I don't know where $\lambda$ is coming from.**
> >
> > **A6.**  Thanks for your suggestions! We are sorry about the confusion. $\lambda$ is the weight to control the KL divergence in Eq (5) and we provide detailed derivation about Eq (12) in Appendix B.2. We have added the demonstration of $\lambda$ below Eq (12).
> >
> > **Q7.  It's not clear how DSPO incorporates $\mathbf{x}_t^w$.**
> >
> > **A7.** Thanks for your comments! We are sorry about the confusion. For simplicity of notation, we use $\mathbf{x}_t$ in place of $\mathbf{x}_t^w$ in the method section. We have added this clarification in the preliminary section.
> >
> > **Q8. SD1.5 is a relatively old model. I request that the authors at least consider SDXL experiments.**
> >
> > **A8.** Thanks for your suggestions! We provide the corresponding results in A2 and A1.
> >
> > **Q9. Are there any sample-efficient aspects of DSPO?**
> >
> > **A9.** Thanks for your questions! It’s a very interesting thought. We randomly select 10% of the Pick-a-Pic data to use DSPO to fine-tune SD15 (we denote DSPO trained on the 10% of the original data as DSPO-part ), resulting in the following outcomes:
> >
> > |     |       |       |PickV2   |       |       |
> > |-----------|----------|-------|-------|-------|-------|
> > |           | Pick Score  | HPS  | Aesthetics | CLIP | Image Reward |
> > | Diffusion-DPO   |  71.60 | 70.20 | 66.20 | 58.80 | 63.60 |
> > | DSPO-part   |  64.80 | 79.00 | 72.40 | 58.80 | 72.40 |
> > | DSPO   |   **73.60**  | **84.80**  | **76.20** | **61.80** | **78.00**  |
> >
> >
> > |     |       |       |Parti-Prompt  |       |       |
> > |-----------|----------|-------|-------|-------|-------|
> > |           | Pick Score  | HPS  | Aesthetics | CLIP | Image Reward |
> > | Diffusion-DPO   | 61.18  | 66.48  | 60.42 | 58.80 |  63.60 |
> > | DSPO-part   |  58.73  | 73.57  | 68.82 | 50.61 |  65.68 |
> > | DSPO   |  **65.32**  | **87.50**  | **76.96**  | **54.86** | **71.75** |
> >
> >
> > |     |       |       |HPSV2  |       |       |
> > |-----------|----------|-------|-------|-------|-------|
> > |           | Pick Score  | HPS  | Aesthetics | CLIP | Image Reward |
> > | Diffusion-DPO   | 76.06   | 72.13  | 66.00 | 58.50 | 64.22   |
> > | DSPO-part     |  68.01  | 73.81  | 67.62 | 54.91 |  65.63 |
> > | DSPO   | **79.90**   | **92.56**  | **80.59** |  **61.13** | **82.31** |
> >
> > We observe that with only 10% of the samples, our model achieves results comparable to those trained on the full dataset with DSPO when tested on PickV2 and outperforms Diffusion-DPO on most metrics across all three datasets. This can demonstrate the sample efficient aspects of our model.

---

> > > ### Author Response · Authors · 2024-11-18
> > > **Response to Reviewer gpAu (part 3/3)**
> > >
> > > **Q10. The authors could also consider using human-benchmark arenas such as imgsys [2] for evaluation.**
> > >
> > > **A10.** Thanks for your suggestions! We follow the evaluation dataset mostly used in previous preference learning algorithms in the diffusion model in a fair setting. It’s challenging for us to gather sufficient feedback for our model’s evaluation within the limited rebuttal period from the imgsys website. Additionally, we conducted a simple human evaluation following the Diffusion-DPO approach (details provided in A3 of reviewer wk6Q) and **achieved a  67.5% win rate compared to Diffusion-KTO**, the state-of-the-art method (except from DSPO) as reported based on reward models. We will leave using imgsys for future work.
> > >
> > > **Q11. To assess the practical aspects of DSPO, it would be useful to report the wall-clock time and memory requirements of DSP and compare them against the existing methods.**
> > >
> > > **A11.** Thanks for your suggestions! We compare our method with Diffusion-DPO by fine-tuning SD15, using a batch size of 16 and 128 gradient accumulation steps on a single A100 GPU. The table below presents the average time per optimization step for DSPO and DPO:
> > >
> > > |           | GPU Memory | Time for each step |
> > > |-----------|----------|-------|
> > > | Diffusion-DPO   |  60.5G |   4.14min    |
> > > | DSPO   | 60.2G | 4.18min     |
> > >
> > > We observe that our models have similar GPU memory and time for each optimization step compared with Diffusion-DPO.

---

> ### Comment · Reviewer_gpAu · 2024-11-19
>
> Thank you very much for providing detailed responses to my comments. I appreciate it! I have updated my score as well.
>
> Consider this comment as a consolidated response to all the comments you have made in response to my review.
>
> I think the new results are very promising and definitely help to place DSPO as a more favorable alternative to other existing methods. Below are some suggestions in terms of editing your manuscript:
>
> 1. Since you have now gathered the comparative scores for MaPO, I suggest using them in the main results as well. IMO, this will help make DSPO an even stronger alternative.
> 2. Consider including the tables from ablation, deduplication experiments, and memory-wall-clock trade-off either in the main paper or the Appendix.
> 3. If you're using an existing codebase/library to perform your experiments, consider citing them appropriately.
>
> I can notice that Diffusion DSPO is still memory intensive, which is where MaPO shines. To try to mitigate that problem, could you try to do LoRA fine-tuning experiment instead of full fine-tuning of the UNet? If this helps achieve results similar to full fine-tuning, then perhaps LoRA fine-tuning could be made the default choice. This might allow you to fine-tune even larger models like Stable Diffusion 3.

---

> ### Author Response · Authors · 2024-11-20
>
> Thank you for your valuable suggestions and increasing your score! We sincerely appreciate the helpful feedback, which has been useful in further improving our paper. In response, we have made the following modifications in the revised paper:
>
> - We put the table of SDXL compared with Mapo and also include Diffusion-DPO and SFT baselines in Table 2.
> - The memory and wall-clock experiments are included in Table 3, with the corresponding analysis provided in Section 5.2. Additionally, the ablation study and deduplication experiments are detailed in Table 6 of Appendix E.1.
> - Our code is based on the implementation of Diffusion-DPO and we have added the corresponding link in Appendix D.2.
>
> For more details, please refer to the revised paper.
>
>
> We completely agree with your suggestion that using LoRA can reduce memory usage and training time. To explore this, we conducted a quick experiment fine-tuning SD15 with LoRA and obtained the following results:
>
> |     |       |       |PickV2   |       |       |
> |-----------|----------|-------|-------|-------|-------|
> |           | Pick Score  | HPS  | Aesthetics | CLIP | Image Reward |
> | DSPO-lora   | 68.00  | 79.40  |  74.40 |   60.20   | 70.80 |
> | DSPO   |   **73.60**  | **84.80**  | **76.20** | **61.80** | **78.00**  |
>
> We have observed that fine-tuning with LoRA enhances performance, though it slightly falls short of achieving results similar to full parameter training. However, the performance gap between these two methods is minimal, making LoRA a viable approach for fine-tuning large models. Next, we plan to fine-tune DSPO on SD3 with LoRA to evaluate its performance as future work. Thank you again for your valuable advice!

---

> > ### Comment · Reviewer_gpAu · 2024-11-20
> >
> > Thanks for providing further inputs!
> >
> > Interesting to see the LoRA results. I would suggest including those results in the Appendix as it might be appealing for users that may not have the sufficient GPU memory to perform full fine-tuning.

---

> > > ### Author Response · Authors · 2024-11-20
> > >
> > > Thanks for your further response and suggestions! We have updated these results in Table 6 of Appendix E.1. If you have any suggestions or concerns, please don't hesitate to share them or reach out with further questions.

---

### Official Review · Reviewer_wk6Q · 2024-11-01

**Soundness:** 3
**Presentation:** 3
**Contribution:** 3
**Rating:** 6
**Confidence:** 4

**Summary:**

This paper proposes a new direct score preference optimization method for diffusion model alignment that utilizes a target human-preferred score function, thereby aligning the fine-tuning objective with the pretraining objective.

**Strengths:**

- The paper takes a different approach to aligning text-to-image diffusion models, motivated by score matching, which sets this method apart from the others.
- In terms of multiple open-source reward scores, DSPO demonstrates effectiveness in increasing reward values.

**Weaknesses:**

- In general, I find that many claims are too vague and ambiguous. When we examine the final loss of Diffusion-DPO and DSPO, how can we definitively say that one aligns with the pretrained loss of Stable Diffusion more clearly than the other? Additionally, why is aligning the diffusion model with direct reward optimization or RL considered suboptimal due to a mismatch in pretraining and fine-tuning objectives? Is there any theoretical justification beyond the win rates?
- In my opinion, since the method is based on human preference, a human evaluation should be conducted to confirm whether it truly increases the reward aligned with human preference. Relying solely on open-source reward model scores seems unreliable, as these models can carry inherent biases.
- Furthermore, why does the Diffusion-KTO result differ so significantly from the original paper? I think the authors should provide detailed explanations of their evaluation settings, including the seeds used, the number of images generated per method, and other relevant factors. Without this information, the results may appear unreliable.

**Questions:**

- In Figure 2, what baseline models were used to calculate the win rate? Is it the pretrained SD1.5 model?
- During evaluation, did the authors generate images from multiple fixed seeds and average the results over them, or do the results come from a single specific seed?

---

> ### Author Response · Authors · 2024-11-18
> **Response to Reviewer wk6Q (part 1/2)**
>
> We gratefully appreciate your time in reviewing our paper. We would like to clarify some misunderstandings regarding our approach.
>
> **Q1. In general, I find that many claims are too vague and ambiguous. When we examine the final loss of Diffusion-DPO and DSPO, how can we definitively say that one aligns with the pretrained loss of Stable Diffusion more clearly than the other?**
>
> **A1.** Thanks for your questions! The Diffusion-DPO loss differs from the pretraining loss used in Stable Diffusion models. Specifically, in the pretraining stage of diffusion models, **the objective relies on denoising objectives or score matching objectives**, as demonstrated in Eq (7) in [1], e.g. $\|\mathbf{s}\_{\theta}(\mathbf{x}(t), t)-\nabla_{\mathbf{x}(t)} \log p_{0 t}(\mathbf{x}(t) \mid \mathbf{x}(0))\|_2^2$. However, Diffusion-DPO takes a different approach by **using a log-likelihood objective to train a BT-model classifier** that distinguishes between preferred and dispreferred images as shown in Eq (7) of our paper. **Therefore, the objective of a classification loss is different from score matching.** Our proposed DSPO method, on the other hand, applies the score matching objective for fine-tuning, aligning with the pretraining loss approach. This method fine-tunes diffusion models to match human-preferred scores as shown in Eq (10), maintaining consistency with the original score matching objective in the pretraining stage.
>
>
> **Q2. Additionally, why is aligning the diffusion model with direct reward optimization or RL considered suboptimal due to a mismatch in pretraining and fine-tuning objectives? Is there any theoretical justification beyond the win rates?**
>
> **A2.** Thank you for your comments! We apologize for any confusion. In Figure 2, we empirically demonstrate the suboptimality of DPO-based algorithms for diffusion models. To further clarify this, we provide the following reasons why DPO-based algorithms for diffusion models are suboptimal compared to our method, DSPO:
>
> - The actual loss of Diffusion-DPO, adapted from the LLM domain, **acts as an upper bound on the direct preference optimization (DPO) loss, as shown in Eq. (12) of [1]. Consequently, optimizing this upper bound may lead to suboptimal results compared with the actual loss.** And Diffusion-KTO builds upon Diffusion-DPO and retains similar drawbacks. In contrast, our DSPO is equivalent to optimizing the RL loss without relying on any such upper bound, as established in Theorem 1.
>
> - **DPO can be interpreted as the forward KL divergence between the learnable policy network and the optimal policy network, which may lead to suboptimal results**, as shown in Figure 1, Theorem 3.2, and Theorem 3.3 in [2]. In contrast, DSPO optimizes the reverse KL divergence similar to RL objectives, as demonstrated in Eq. (15) and Eq. (16) in our paper. This approach encourages a mode-seeking policy, **allowing it to capture key characteristics of the optimal policy more effectively and perform better than DPO, as discussed in [2].**
> - Empirical experiments demonstrate that DSPO also outperforms direct reward optimization methods on diffusion models.
>
> Furthermore, DSPO **introduces a novel perspective by designing a preference learning loss derived from the diffusion loss itself (score matching).** Rather than simply adopting methods from the LLM domain, our approach aims to inspire fresh ideas and innovation within this field.
>
>
> **Q3. In my opinion, since the method is based on human preference, a human evaluation should be conducted to confirm whether it truly increases the reward aligned with human preference**
>
> **A3.** Thanks for your suggestions! Following the human evaluation methodology in Diffusion-DPO, we use prompt indices 200, 600, 1000, 1400, 1800, 2200, 2600, and 3000 from the HPSV2 dataset. Five labelers assess these prompts by selecting their preferred images. The results below compare our DSPO method with Diffusion-KTO, the state-of-the-art method (except from DSPO) as reported based on reward models. **DSPO achieves a 67.5% win rate compared with Diffusion-KTO.** This further verifies the effectiveness of our method.

---

> > ### Author Response · Authors · 2024-11-18
> > **Response to Reviewer wk6Q (part 2/2)**
> >
> > **Q4. Furthermore, why does the Diffusion-KTO result differ so significantly from the original paper?**
> >
> > **A4.** Thanks for your comments! Different baselines, such as Diffusion-DPO and Diffusion-KTO, use varying guidance scales for their diffusion models. In our experiments, we adhere to the settings from Diffusion-DPO, utilizing a default guidance scale of 7.5 and 50 sampling steps for all trials (we demonstrate it in line 48). We are not sure about these hyperparameter settings , pretrained reward models and evaluation code in Diffusion-KTO. **For example, aesthetic reward model includes various trained versions through [link1](https://github.com/LAION-AI/aesthetic-predictor) or [link2](https://github.com/SalesforceAIResearch/DiffusionDPO/blob/main/utils/aes_utils.py) and we use the evaluation model provided in Diffusion-DPO by [link](https://github.com/SalesforceAIResearch/DiffusionDPO) including Pick Score, HPS, Aesthetics, CLIP. And we use ImageReward-v1.0 used in [link](https://github.com/THUDM/ImageReward).**
> >
> > In our experimental setup, we select one random seed 0 (seed used in Diffusion-DPO) and randomly select other 4 seeds  to ensure consistency across methods and to compute averaged results from these 5 seeds in Table 1,2,3 and Figure 2,4 and 7. All quantitative results presented in this paper are based on experiments conducted with seed 0. We will release all code to reproduce the results after paper acceptance.
> >
> > **Q5. In Figure 2, what baseline models were used to calculate the win rate? Is it the pretrained SD1.5 model?**
> >
> > **A5.** Thanks for your comments! We use the SD15 as the base model to calculate the win rate and have added this clarification in the revised paper.
> >
> > **Q6. Did the authors generate images from multiple fixed seeds and average the results over them, or do the results come from a single specific seed?**
> >
> > **A6.** Thanks for your question! Yes, we generate images with five seeds for all baselines and average the results.
> >
> > [1] Diffusion Model Alignment Using Direct Preference Optimization.
> >
> > [2] Towards Efficient Exact Optimization of Language Model Alignment.

---

> > > ### Author Response · Authors · 2024-11-20
> > > **A sincere and kind reminder to the Reviewer wk6Q: would you mind confirming if our rebuttal addresses your comments?**
> > >
> > > Dear ICLR Reviewer wk6Q,
> > >
> > > We gratefully appreciate your time in reviewing our paper and your comments.
> > >
> > > We have made extensive efforts to address your comments and believe that they adequately address all your concerns. The reviewer's comments are mainly about some clarifications and are not fatal to the contributions of our manuscript; we believe that the reviewer's insightful comments can be easily and effectively addressed in the final version.
> > >
> > > We would like to confirm whether there are any other clarifications they would like. If the reviewer's concerns are clarified, we would be grateful if the reviewer could increase the score.
> > >
> > > Many thanks for your time; we are extremely grateful.
> > >
> > > The authors of “DSPO: Direct Score Preference Optimization for Diffusion Model Alignment”

---

> ### Author Response · Authors · 2024-11-23
> **Kind reminder to Reviewer wk6Q: discussion period is ending soon; we would like to hear back from Reviewer wk6Q**
>
> Dear ICLR Reviewer wk6Q,
>
> We greatly appreciate your time and the insightful comments provided during the review of our paper.
>
> We have made extensive efforts to address all your questions, suggestions, and misunderstandings in the response and believe that they adequately address all your concerns. The reviewer's comments primarily focused on clarifying certain claims and experimental details. We have addressed these points in our response by providing detailed explanations, including clarifications on specific claims, experimental procedures, and the results of human evaluation experiments. We believe that the reviewer's insightful comments can be easily and effectively addressed in the final version.
>
> With the public discussion phase ending soon, we would like to confirm whether there are any other clarifications they would like. We would be grateful if the reviewer could increase the score.
>
> Many thanks for your time; we are extremely grateful.
>
> Best regards,
>
> The authors of “DSPO: Direct Score Preference Optimization for Diffusion Model Alignment”

---

> ### Comment · Reviewer_wk6Q · 2024-11-25
> **Response to Rebuttal**
>
> I apologize for thelate reply. Thank you so much for your thorough rebuttal addressing my concerns.
>
> Most of my concerns, especially regarding human evaluations and a few minor clarifications, have been resolved.
>
> However, I still believe that if we focus solely on the final DPO loss, without considering the intermediate derivations, the process essentially predicts the noise better for the chosen sample compared to the reference model, and the opposite for the not-chosen sample. I don't see this as a critical mismatch with the pretraining loss. Additionally, why is it so important to have the same objective for pretraining and fine-tuning?
> But it is also true that incorporating score-matching loss is an interesting direction.
>
> Taking these points into account, I will raise my score to 6. Once again, thank you for your hard work.

---

> > ### Author Response · Authors · 2024-11-25
> >
> > Thank you for your insightful questions and feedback! We sincerely appreciate your consideration in raising your score. We apologize for any confusion caused by our earlier phrasing. We fully agree with your point that the mismatch is not a definitive conclusion, nor can we state with certainty that this mismatch directly leads to suboptimal results.
> >
> > In the revised paper, we have clarified this in the introduction. Specifically, we now analyze the optimality of results based on the estimated loss of Diffusion-DPO (as an upper bound of the DPO loss in LLM domains) and empirical studies. We also emphasize that our proposed algorithms offer a fresh perspective on this research direction.
> >
> > Once again, we sincerely thank you for your thoughtful questions, which have been invaluable in helping us improve our paper.

---

### Official Review · Reviewer_HsUR · 2024-11-23

**Soundness:** 4
**Presentation:** 3
**Contribution:** 4
**Rating:** 8
**Confidence:** 2

**Summary:**

The authors propose the first method to fine-tune text-to-image models on human preference that aligns pertaining and fine-tuning objectives, the benefit of which is shown by outperforming preference learning baselines on human preference tasks. The method is also shown to be equivalent with RLHF objectives in diffusion models under certain conditions.

**Strengths:**

1. The first method that fine-tunes diffusion models on human preferences using a score-matching, thus aligning with pretraining objectives, and the benefits are apparent through the baselines.
2. Clear presentation of related works provides the reader with ample context.
3. Comprehensive experiments covering supervised finetuning, diffusion-DPO, diffusion-KTO, and MaPO as baselines, evaluated with multiple scoring metrics, and includes evaluations on a recent model (SDXL).
4. The proposed method outperforms baseline methods across multiple datasets and scoring metrics.

**Weaknesses:**

1. Figure 3 looks cramped; the equations are pitted against other visual elements, making them hard to read

**Questions:**

1. In table 2, Diff.-DPO seems to have a higher score in the PickV2 dataset with the Pick Score Metric, but it’s not highlighted.
2. The paper shows the equivalence between the proposed method and RLHF, but it seems like the evaluation does not contain a direct comparison with a model that has been fine-tuned with RLHF.

---

> ### Author Response · Authors · 2024-11-23
>
> Thank you for the great summarization of our contributions, and we really appreciate your encouraging comments. Please see our responses below:
>
> **Q1. Figure 3 looks cramped; the equations are pitted against other visual elements, making them hard to read.**
>
> **A1.** Thank you for your valuable suggestions! We have updated the figure to enhance its clarity, and you can find the revised version in our updated paper.
>
> **Q2. In table 2, Diff.-DPO seems to have a higher score in the PickV2 dataset with the Pick Score Metric, but it’s not highlighted.**
>
> **A2.** Thank you for pointing this out! We have emphasized the Pick Score for PickV2 in the revised paper, and you can review the updates there.
>
> **Q3. The paper shows the equivalence between the proposed method and RLHF, but it seems like the evaluation does not contain a direct comparison with a model that has been fine-tuned with RLHF.**
>
> **A3.** Thanks for your insightful questions! Our paper focuses on preference learning for human alignment, utilizing only human preference data without reward models, such as pairs of preferred and non-preferred images [1]. Our model is fine-tuned on this offline dataset. In contrast, RLHF algorithms firstly require reward models and fine-tune their models by sampling from a policy network and utilizing reward models, operating under fundamentally different settings from our approach. This distinction makes it challenging to conduct a fair comparison between our DSPO and RLHF algorithms. Moreover, the optimization of the two stages in RLHF algorithms adds complexity to the process and results in significant training costs, even for larger datasets [1],[2],[3]. We have also included a state-of-the-art baseline for preference learning in diffusion models that shares the same setting as our algorithm. This has demonstrated the effectiveness of our proposed DSPO.
>
> [1] Diffusion Model Alignment Using Direct Preference Optimization
>
> [2] Direct Preference Optimization: Your Language Model is Secretly a Reward Model
>
> [3] Aligning Diffusion Models by Optimizing Human Utility

---

### Meta-Review · Area_Chair_YQ4n · 2024-12-19

**Metareview:**

This paper proposed a method to fine-tune diffusion models on human preferences using score-matching (and hence its objective will be the same as pre-trained models).

For paper strength, technical novelty (e.g., the first finetuning method with score matching), solid experiments (comprehensive experiments against baselines like diffusion-DPO, diffusion-KTO, and MaPO), and theoretical analysis (about the connection between RLHF and the proposed DSPO) are the major advantages of this paper.

I feel this paper has a novel idea for an important direction which is well motivated, and provided both solid experiments and theoretical analysis to justify the proposed method. So I will recommend "Accept" for this paper.

**Additional Comments On Reviewer Discussion:**

Main weaknesses of the paper are: insufficient clarity of some technical part, as well as lack of some experiments/results, e.g., with latest diffusion model like SDXL, or evaluation with human study. The authors have addressed most of the issues in the rebuttal.

---

### Decision · Program_Chairs · 2025-01-22

Accept (Oral)